# Sequential order dependent dark-exciton modulation in bi-layered TMD heterostructure

Riya Sebait[1,2], Roberto Rosati[3], Seok Joon Yun [4], Krishna P. Dhakal[1], Samuel Brem [3], Chandan Biswas [2], Alexander Puretzky [4], Ermin Malic [3] ✉ & Young Hee Lee [1,2] ✉

We report the emergence of dark-excitons in transition-metal-dichalcogenide (TMD) heterostructures that strongly rely on the stacking sequence, i.e., momentum-dark K-Q exciton located exclusively at the top layer of the heterostructure. The feature stems from band renormalization and is distinct from those of typical neutral excitons or trions, regardless of materials, substrates, and even homogeneous bilayers, which is further confirmed by scanning tunneling spectroscopy. To understand the unusual stacking sequence, we introduce the excitonic Elliot formula by imposing strain exclusively on the top layer that could be a consequence of the stacking process. We further find that the intensity ratio of Q- to K-excitons in the same layer is inversely proportional to laser power, unlike for conventional K-K excitons. This can be a metric for engineering the intensity of dark K-Q excitons in TMD heterostructures, which could be useful for optical power switches in solar panels.

Monolayer transition-metal-dichalcogenide (TMD) semiconductors have a direct bandgap and exhibit optically accessible bright excitons[1–4]. In addition, they also show a variety of optically forbidden dark excitons, either due to spin-flip or momentum transfer[5–7]. For example, both K-K bright and momentum-forbidden K-Q excitons emerge in monolayer WSe$_2$, as confirmed by time- and angle-resolved photoemission spectroscopy[6]. These momentum-forbidden indirect excitons can also be realized in van der Waals (vdW) heterostructures, where bound electrons and holes are localized in two different layers, which is called interlayer excitons[7–14]. Emergence of these indirect interlayer excitons strongly depends on the stacking angle between the layers[8,10,12] and is independent of stacking sequential order[10,15]. Despite this, a few recent reports have accidentally observed optical anomalies in heterostructures that rely on the sequential order of the layers[15,16]. However, the underlying mechanisms that give rise to these anomalies are not yet fully realized. This lack of understanding may be attributed in part to the hypothesis that altering the stacking sequence between layers would not significantly affect the intrinsic properties of heterostructures. Therefore, it is important to explore stacking sequential order dependence to further use of real device applications.

In this work, we tune dark excitons by changing the stacking sequence between TMD layers, whereas other experimental parameters remain constant. Such emergence or disappearance of dark excitons depending upon stacking sequence has not been reported to date. Moreover, the emergence of intra-layer dark exciton in TMDs heterostructure also remains elusive. This phenomenon is investigated by analyzing the photoluminescence (PL) spectra of 2D TMD heterostructures in different stacking configurations with a series of experiments to reveal such an unusual phenomenon. Furthermore, our microscopic theory with the generalized Elliot formula suggests that an additional strain on the top layer stemming from the fabrication process could be a solution to explain such an unusual phenomenon.

[1]Deparment of Energy Science (DOES), Sungkyunkwan University, Suwon 16419, Republic of Korea. [2]Center for Integrated Nanostructure Physics (CINAP), Institute for Basic Science (IBS), Sungkyunkwan University, Suwon 16419, Republic of Korea. [3]Department of Physics, Philipps-Universität Marburg, Marburg 35032, Germany. [4]Center for Nanophase Materials Sciences (CNMS), Oak Ridge National Laboratory, Oak Ridge, TN 37830, USA. ✉e-mail: ermin.malic@physik.uni-marburg.de; leeyoung@skku.edu

## Results and discussion

### Stacking sequence under various circumstances

We investigate the stacking sequence of bilayers at various conditions: heterogeneous TMDs (WSe$_2$/WS$_2$ and WS$_2$/MoSe$_2$), different substrates (SiO$_2$ and hBN), and homogeneous TMDs (WSe$_2$/WSe$_2$ and WS$_2$/WS$_2$) (Fig. 1). In each case, one layer (bottom layer) has been exfoliated mechanically on the substrate. Another layer (top layer) has been exfoliated on top of the PMMA-coated substrate, which is then dry transferred to the aforementioned layer (see methods and Supplementary Fig. 1). Individual monolayers, as well as the vertical heterostructures were confirmed by Raman spectra (Supplementary Fig. 2) and the height profile in atomic force microscopy (AFM) (Supplementary Fig. 3). The interface of the stacked bilayer has not been touched by contaminants, such as PMMA, during dry transfer to ensure a clean interface (Supplementary Note 3).

Figure 1a, b represents a schematic for vertical TMD heterostructures of top-WS$_2$/bottom-WSe$_2$ and top-WSe$_2$/bottom-WS$_2$ on SiO$_2$/Si substrate together with optical images (inset). We find clear PL peaks appearing around 2.0 eV for the WS$_2$ monolayer (green dotted lines) and around 1.64 eV for the WSe$_2$ monolayer (blue dotted lines), corresponding to the A-excitons of the layers. An additional feature (marked as S$_Q$) emerges near 1.89 eV below the A-exciton peak of the top layer (WS$_2$) at WS$_2$/WSe$_2$ heterostructure (Fig. 1a). Meanwhile, in

the reverse stacking sequence of the WSe$_2$/WS$_2$ heterostructure, another feature (marked as Se$_Q$) emerges around 1.56 eV below the A-exciton of the top WSe$_2$ layer, whereas the previous feature (S$_Q$) near the WS$_2$ layer notably disappears (Fig. 1b and Supplementary Fig. 8). The presence of such additional peaks close to the top layer is well distinguished from trions or bi-exciton peaks, as shown in the deconvoluted curves (Fig. 1a left, 1b, right and Supplementary Fig. 9)[8,17]. This unusual stacking sequence has not been reported to date. We confirmed the reproducibility of such additional peaks in multiple heterostructures (a total of 35) for both stacking sequences (Supplementary Fig. 10). We note that all heterostructures were randomly stacked and the stacking angle between the layers was further confirmed by second harmonic generation (SHG) measurements (Supplementary note 7). Therefore, the observed features are not specific to a stacking angle. This suggests that our additional peaks originate neither from interlayer excitons[8,10] nor moiré excitons[17,18], as they require precise angle alignment between the layers. Furthermore, the energies of these additional peaks (1.89 and 1.56 eV) are distinct from reported interlayer excitons (1.35–1.42 eV) for the WS$_2$-WSe$_2$ heterostructure[8,17]. Moreover, we observed weak interlayer exciton peak at an energy of 1.35 eV in MoSe$_2$/WSe$_2$ heterostructure at room temperature (Supplementary Fig. 12). This is well matched with previously reported results[10,12,19], which is rarely observed as it is contingent upon a specific stacking angle.

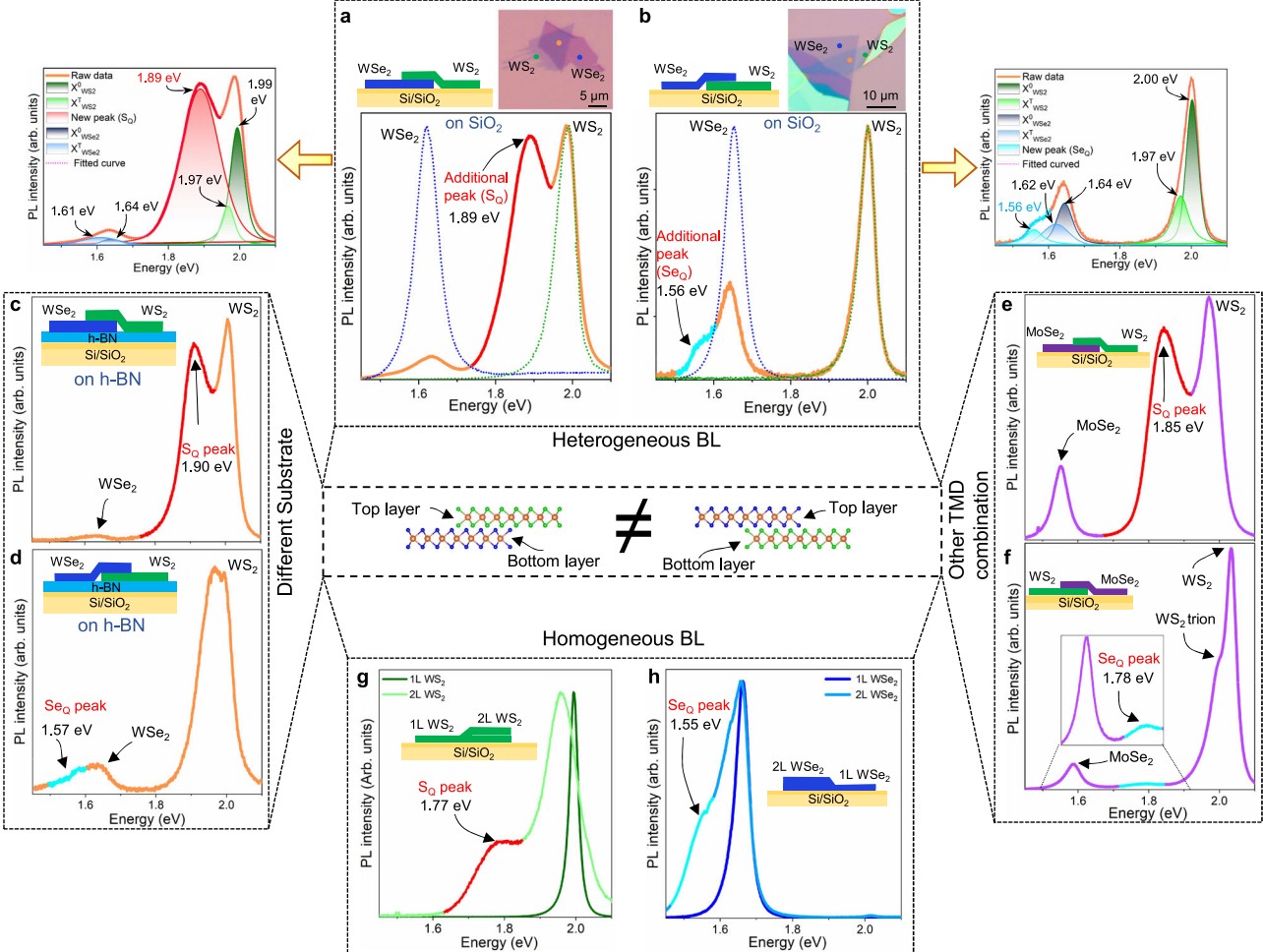

**Fig. 1 | Dark-exciton modulation in TMDs heterostructures. a** Schematic along with optical images of WS$_2$/WSe$_2$ heterostructure on SiO$_2$/Si substrate. PL spectra of individual monolayer (dotted lines) with heterostructure (solid line); A red color peak at heterostructure indicates a dark exciton peak (S$_Q$) near the A-exciton peak of WS$_2$, which is deconvoluted for detailed analysis (left). **b** Similar for the inverted stacking sequence of WSe$_2$/WS$_2$. Dark exciton peak (Se$_Q$) emerges near A-exciton peak of the top WSe$_2$, in stark contrast with that in **a**. **c, d** The effect of different substrates with h-BN. Dark exciton peaks (S$_Q$ and Se$_Q$) are still persistent with corresponding stacking sequences. **e, f** The effect of different materials in combination with WS$_2$-MoSe$_2$. Dark exciton peaks still appear near the top layer. **g, h** Emergence of dark exciton peaks at homogeneous bilayers of each WS$_2$ and WSe$_2$.

We now consider the substrate effect involving possible strains, charge traps, and dielectric screening by fabricating both heterostructures on the top of h-BN layer (Fig. 1c, d). Both features ($S_Q$ and $Se_Q$) are still visible for the corresponding stacking sequences, although the peak positions were slightly upshifted (Supplementary Fig. 13). The $S_Q$ peak is also visible with other substrates including quartz and $HfO_2$ (Supplementary Fig. 14). We further confirm the existence of these peaks by encapsulating both sides of heterostructures by h-BN as well as $SiO_2$ to exclude possible environmental effects (Supplementary Fig. 15). These results confirm that the additional features are independent of the substrate, which could be related to the strong interfacial coupling of the hetero-bilayer. We also investigate the stacking sequence with other TMDs combinations (Fig. 1e, f). The $S_Q$ peak still appears around 1.85 eV in the $WS_2/MoSe_2$ heterostructure (Fig. 1e). Meanwhile, a similar additional peak ($Se_Q$) emerges in the $MoSe_2/WS_2$ heterostructure (Fig. 1f), but the energy is upshifted, different from the downshifted energy in W-based materials. We further investigated several hetero-bilayers with other types of TMDs, for example, the $MoS_2$-$WSe_2$ or $MoS_2$-$WS_2$ heterostructure (Supplementary Note 10). All these bilayer heterostructures clearly revealed the emergence of additional features near the top layer.

To investigate if the additional peak appears only in heterostructures, we also study the PL in homogeneous bilayers by exfoliating the intact bilayer. The homogeneous bilayer $WS_2$ still clearly shows the $S_Q$ peak, which is well distinguished from the monolayer $WS_2$ (Fig. 1g). Similarly, the $Se_Q$ peak is also observed in bilayer $WSe_2$ (Fig. 1h). This implies that the additional peak does not necessarily originate from the nature of the hetero-bilayer but is inherent from the homogeneous bilayer, congruent with previous reports[20-22]. Furthermore, the exfoliated bilayer interface remains intact and hence the presence of such an unusual peak reassures the clean interface in our fabricated hetero-bilayers (Supplementary Fig. 4). We also observed strong and distinct $S_Q$ peak on the spatially uniform and bubble-free flat heterostructure surface (Supplementary Fig. 17), which differs from previously reported localized exciton due to the presence of bubbles[23].

Additionally, monolayer (or any odd number of layers) TMD has broken inversion symmetry, whereas bi-layer (or any even number of layers) preserved inversion symmetry[24,25]. We have also checked how the broken inversion symmetry affects the additional features ($S_Q$ or $Se_Q$). We first measured the layer-dependent (even and odd layer) PL at homogeneous $WS_2$ as well as $WSe_2$ (Supplementary Fig. 18). The additional feature was still present, except for further downshifting as the number of layers increased, as observed previously[26]. We further checked the heterogeneous heterostructure with the different number of layers, which follows the similar stacking sequence as observed in bilayer heterostructure (see details in Supplementary note 13). We further confirmed similar features $S_Q$ or $Se_Q$ depending on the corresponding stacking sequence at low temperatures (77 K) (Supplementary Fig. 20). These additional peaks were still observed after annealing the sample at 250 °C for 12 h in helium environment (Supplementary Fig. 21). Additionally, we detected similar peaks ($S_Q$ or $Se_Q$) on the doped materials such as Re-doped $WS_2$ and Nb-doped $WSe_2$ samples (Supplementary Fig. 22). All of these observations reassure these additional peaks to be intrinsic, but not related to defect states.

## Q-band downshifts via band renormalization

The emergence of these unusual peaks in PL can be explained in terms of the Q-band downshift in the top layer via band renormalization in the heterostructure region, which has been demonstrated previously for $WS_2$-$WSe_2$ hetero-bilayer[8] and bilayer of $MoSe_2$[27]. In contrast, such peaks ($S_Q/Se_Q$) was absent in absorption measurements due to the indirect nature of $S_Q$ or $Se_Q$ peak (see details in Supplementary Note 15). The band structure of individual monolayer $WS_2$ or $WSe_2$ has K- and Q-bands, as illustrated in Fig. 2a, b resulting in bright K-K and

indirect dark K-Q excitons[6]. We now propose that dark excitons become active and visible due to the downshifted Q-band at the heterobilayer region. For example, in $WS_2/WSe_2$ heterostructure (Fig. 2c), the Q-band is downshifted compared to K-band in the top $WS_2$ layer in the heterostructure region. The momentum indirect phonon-assisted emission from intralayer K-Q exciton becomes stronger by accumulated electron population at the Q-band transferred from the K-band in the same $WS_2$ layer as well as additional electron population transferred from $WSe_2$ to $WS_2$ layer due to type-II band alignment in the heterostructure (Supplementary Fig. 24a, b)[8,28,29]. Additionally, the reduction of relaxation time in heterostructure, measured by time-resolved photolumenesence, can be attributed to fast charge transfer facilitated by both direct K-K band and further indirect K-Q band transition (Supplementary Fig. 24c). This gives rise to strong emission of the $S_Q$ peak. Similarly, for the opposite stacking, K-Q excitons become active in the top $WSe_2$ layer by band renormalization. Furthermore, the K-Q population is less significant in the bottom $WS_2$ layer (Fig. 2d), consequently yielding a relatively low $Se_Q$ intensity. The microscopic origin for the stacking sequence dependence of the additional peaks will be discussed further below and is ascribed to strain appearing in the upper layer during the stacking procedure.

## Electronic band structure

To confirm the Q-band downshift, we examine the position of Q-band in the top-$WS_2$ layer in $WS_2/WSe_2$ heterostructure via scanning tunneling spectroscopy (STS). The schematic of $WS_2/WSe_2$ heterostructure on $SiO_2/Si$ substrate (top) and the optical image of the device (bottom) are shown in Fig. 2e. The dI/dV clearly reveals the additional peak (red) near the conduction band edge in the heterostructure (Fig. 2f), resembling PL peaks in Fig. 1. This is well contrasted with the simple band profiles from individual $WS_2$ or $WSe_2$ layer (Fermi level is shifted to the middle of the band edges and see Supplementary Note 18).

## Power-dependent PL measurements

To elucidate the underlying mechanism of the seemingly inconsistent stacking sequence, we measure the laser (532 nm) power-dependent PL of the stacking sequence of $WSe_2/WS_2$ heterostructure on the $SiO_2/Si$ substrate (Fig. 3a). As the power rises, the $Se_Q$ peak intensity ($I_Q$) in the top $WSe_2$ layer decreases compared to the A-exciton peak ($I_K$) (Fig. 3a, right). This reduction of the $Se_Q$ intensity is ascribed to the charge screening of saturated electron density in the Q-band at high laser power. Variances of neutral exciton peak ($X^0$) and emerging trion peak ($X^T$) at high power region are not appreciable (Fig. 3b). This tendency holds true for $Se_Q$ peak. At higher power above ~130 $\mu$W, the trion state of $Se_Q$ ($Se_Q^T$) emerges. We observe two screening regions in the power-law dependence with $I = P^\alpha$ where I is intensity, p is power and $\alpha$ is an exponent. For the neutral exciton, the exponent $\alpha_1 = 0.88$, steady in the low power region, and $\alpha_2 = 0.65$, saturated in high power region due to conversion of the neutral exciton to trion[30]. This is again confirmed by the emergence of the trion peaks with a single exponent $\alpha_1 = 0.62$ in the high power region. The $Se_Q$ peak reveals the screening behavior with similar two exponents like neutral exciton in $WSe_2$, except a much lower exponent ($\alpha_2$) in high power region due to the high carrier screening at the Q-band compared to the K-band ($X^0$). This similar power exponent reassures that the nature of the $Se_Q$ peak originates from the intra-band exciton (K-Q).

We next study the power-dependence in the $WS_2/WSe_2$ heterostructure on the hBN substrate (Fig. 3d) as well as on the $SiO_2$ substrate (Supplementary Fig. 27). The $S_Q$ peak appears in the top $WS_2$ layer, which is still persistent with increasing power. The intensity ratio of $I_Q/I_K$ decreases, yielding a similar tendency to the $WSe_2/WS_2$ heterostructure, independent of the substrate of hBN or $SiO_2$ related to trap-charges (Fig. 3d, right)[30]. The $X^T$ peak in the $WS_2$ layer emerges from

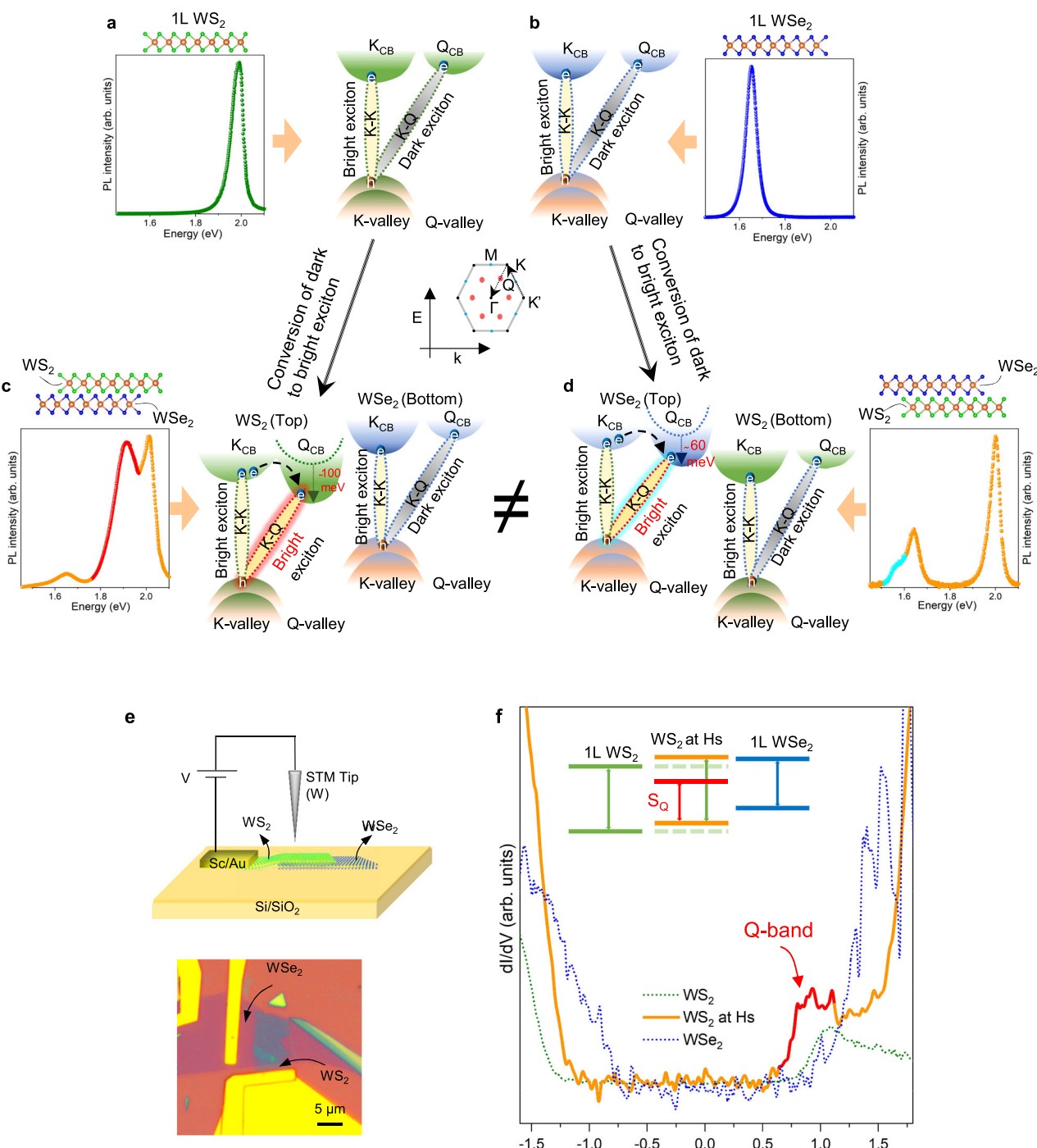

**Fig. 2 | Band renormalization at heterostructure. a, b** Schematic band diagram of individual monolayer $WS_2$ and $WSe_2$ along with the corresponding PL spectrum, where bright exciton appears in K-band, unlike dark exciton in Q-band.
**c, d** Schematic band alignment at $WS_2/WSe_2$ ($WSe_2/WS_2$) heterostructure, where Q-band is downshifted at the top layer, converting dark exciton into bright exciton.

**e** Schematic and optical image of the device for STS measurements. **f** STS profiles at individual monolayer $WS_2$ (green dotted line), $WSe_2$ (blue dotted line), and heterostructure (solid line). Mid-gap state $S_Q$ (red) emerges closely to CB, due to the downshifting of the Q-band at the heterostructure region.

the beginning of low power region due to enhanced carrier density on the h-BN substrate (Fig. 3e). The presence of both $X^0$ and $S_Q$ peaks at low power again indicates that the $S_Q$ peak originates from the intra-band nature (Fig. 3e, f). In addition, the trion state of the $S_Q$ peak ($S_Q^T$) emerges in the low power region of about $60\,\mu W$ again due to enhanced carrier density, and furthermore, the energy is downshifted due to strong charge screening. The power dependence of exciton peaks is similar to those of the corresponding peaks in the top-$WSe_2$

layer except for the elevated power exponent, which is ascribed again to the enhanced carrier density (Fig. 3f)[30].

## Inhomogeneous distribution of dark excitons

We investigate homogeneity of the K-Q excitons over the sample via PL energy and intensity mapping, as shown in Fig. 4a, b (schematic and optical image in Fig. 4a top and inset). The light-bluish milky region with an energy range of 1.88–1.90 eV is referred to as K-Q excitons

(Fig. 4a), which is inhomogeneously distributed over the entire heterostructure region. For clarity, we plotted the normalized $S_Q$ peak distribution over the heterostructure region (details in Supplementary Note 20), where the $S_Q$ population is dominant at the heterostructure region compared to the intrinsic inlayer $WS_2$ exciton intensity (Fig. 4b). Three PL spectra have been extracted from the representative spots (A, B, C). Most of the light-bluish milky region, for example, marked as region A, represents the dominant Q-band peak with downshifted energy as well as high intensity compared to the K-band (Fig. 4c, top). In region B, the Q-peak intensity is reduced and comparable to the K intensity as well as the PL energy is slightly upshifted (Fig. 4c, middle). Moreover, the Q-peak intensity is quenched further with more energy upshift in region C (Fig. 4c, bottom). Some other spots including low-energy (e.g., red circle) regions are attributed to the artifacts, such as air bubbles and contaminants. Such inhomogeneous energy and intensity of the Q-band can be explained as a metric of coupling strength between layers, $\sigma = \frac{I_Q - I_K}{I_Q + I_K}$. $\sigma$ is positive for strong coupling and negative for weak coupling and $\sigma \approx 0$ for medium strength (Fig. 4d).

## Microscopic model for the origin of unusual peaks

The variation of the coupling strength from position to position indicates possible inhomogeneous strain in the bilayered heterostructure. $E_{2g}$ peak shift from Raman spectra could be monitored as a metric for strain. For example, the $E_{2g}$ peak was slightly shifted at each layer in the heterostructure compared to the individual layer due to minute local strain but varied at different positions (Supplementary Fig. 29), making it difficult to identify strain at each layer of the heterostructure. However, in practice, the bottom layer was directly exfoliated on the target substrate, which was strongly anchored on the substrate. The top layer was transferred later on to the bottom layer via dry transfer, in which the top layer was anchored and then strained via vdW interaction (see Methods and Supplementary Fig. 30). Yet, the interlayer coupling varies with positions, resulting in different strains (Fig. 4a–c).

We introduce a generalized Elliot formula including indirect phonon-assisted recombination of momentum-dark excitons in our theoretical model for calculating the PL emission of strained TMD monolayers[31]. By applying a small compressive strain on the top layer (top-$WS_2$ layer: 0.3% and top-$WSe_2$ layer: 0.15% strain in heterostructures), we can successfully reproduce the experimental data (Fig. 4e, f) for W-based materials. While the theoretical results shown in Fig. 4e, f are obtained for 77 K for better visualization, a strain-induced increase of the energy separation between K-Q and K-K excitons (Supplementary Fig. 31)[31] could lead to visible phonon-sidebands also at higher temperatures, in accordance to experimental observations of $Se_Q$ and $S_Q$. In contrast, Mo-based material showed K-Q dark exciton higher in energy compared to K-K bright exciton even in the presence of compressive strain (Details in Supplementary Fig. 31). This leads to a negligible phonon-assisted PL. Therefore, the presence of a strained top layer, leads to a new peak in $WS_2/MoSe_2$, corresponding to lower energy K-Q states, while no new phonon-assisted peaks are observed in $MoSe_2/WS_2$ due to higher energy K-Q excitons, resulting in negligible phonon-assisted PL (Supplementary Fig. 32).

To explore this phenomenon further, we measured the laser path-dependent PL for the same sample ($WS_2/WSe_2$ heterostructure) by fabricating our heterostructure on top of the transparent substrate (quartz) and illuminating the laser from the top (bottom) heterostructure (Supplementary Fig. 33). The presence of the $S_Q$ peak from the top $WS_2$ layer (which was not directly attached to the substrate) in both cases, again confirmed that this phenomenon is independent of the optical path.

## Optical power switch

Finally, we investigate the variation of the PL intensity with laser power in the strong coupling region (A), which was demonstrated in Fig. 4g, where $I_Q$ is reduced, and $I_K$ is elevated with increasing laser power. Interestingly, the polarity of $\sigma$ in the strong coupling region is clearly

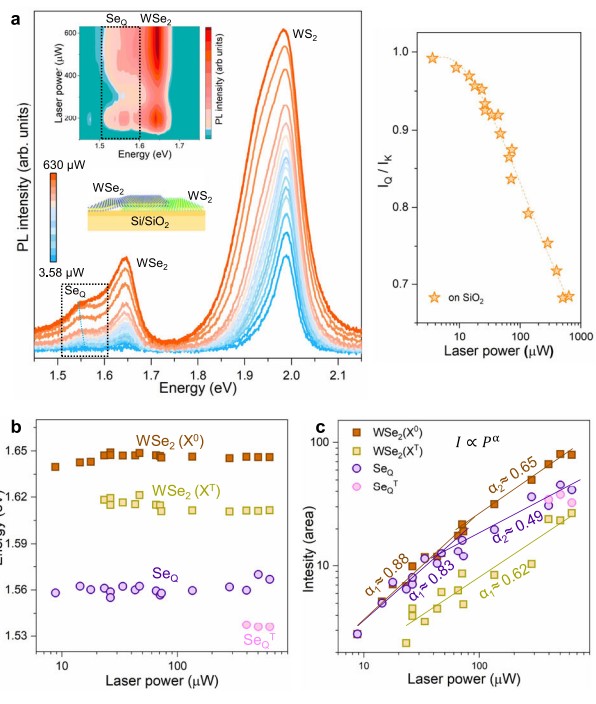

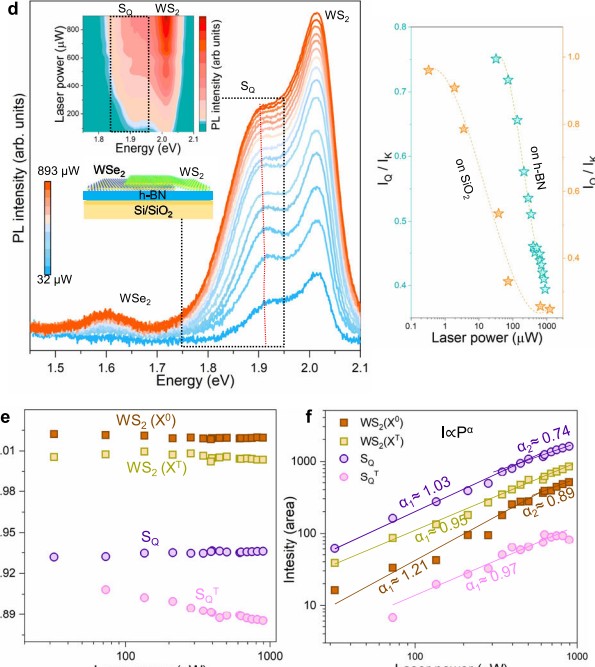

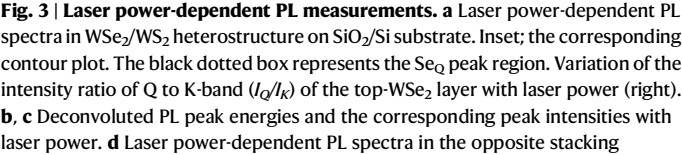

**Fig. 3 | Laser power-dependent PL measurements. a** Laser power-dependent PL spectra in $WSe_2/WS_2$ heterostructure on $SiO_2/Si$ substrate. Inset; the corresponding contour plot. The black dotted box represents the $Se_Q$ peak region. Variation of the intensity ratio of Q to K-band ($I_Q/I_K$) of the top-$WSe_2$ layer with laser power (right). **b, c** Deconvoluted PL peak energies and the corresponding peak intensities with laser power. **d** Laser power-dependent PL spectra in the opposite stacking sequence of $WS_2/WSe_2$ heterostructure on the h-BN substrate. Inset: contour plot of the spectra and the black dotted box represents the $S_Q$ peak region. Variation of $I_Q/I_K$ of the top-$WS_2$ layer with laser power on $SiO_2$ and the h-BN substrate (right). **e, f** Deconvoluted PL peak energies and the corresponding peak intensities of the top-$WS_2$ layer with laser power on the h-BN substrate.

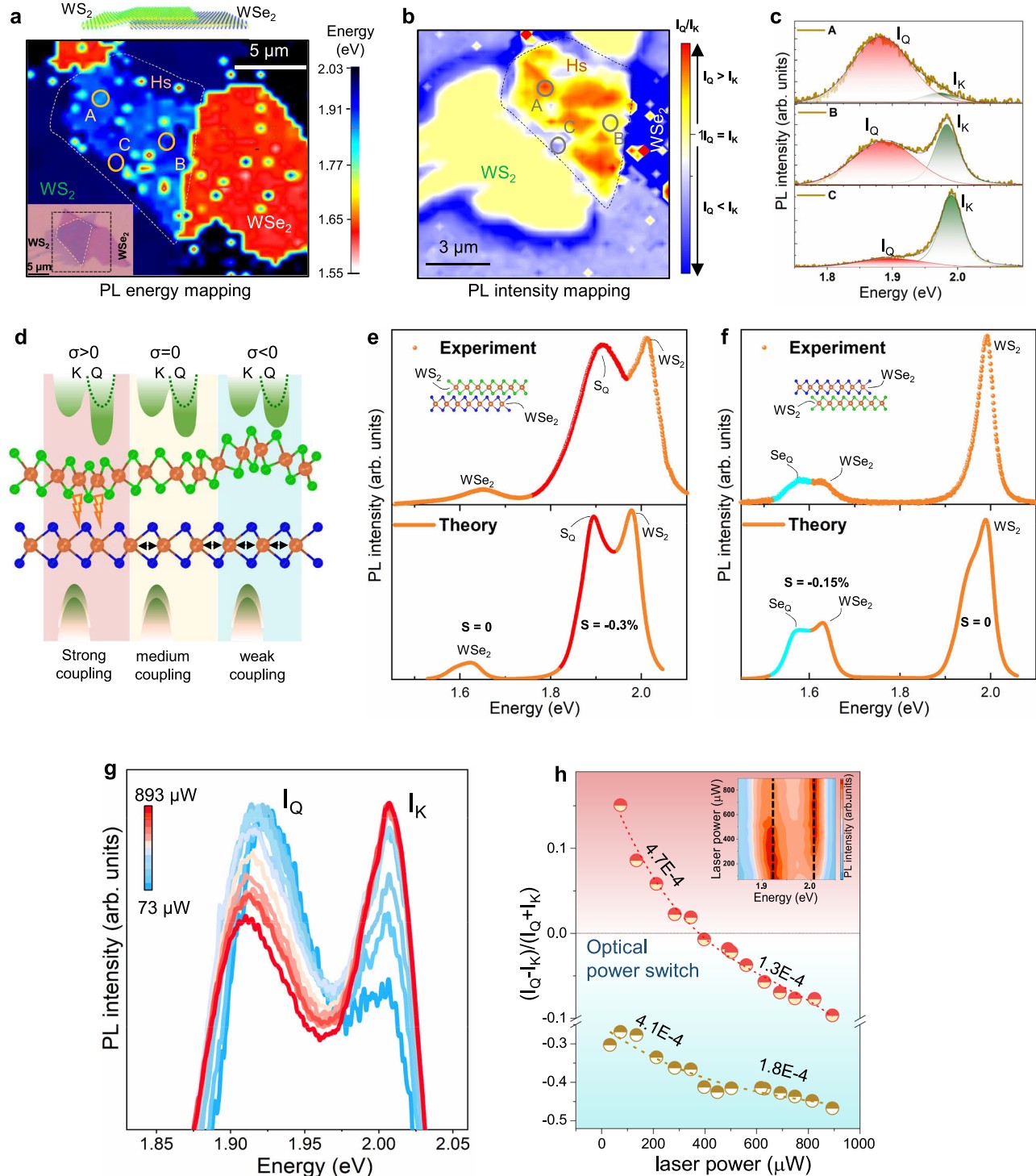

**Fig. 4 | Theoretical modeling and application of Q-exciton as an optical power switch. a, b** PL energy and intensity mapping of WS$_2$/WSe$_2$ heterostructure at the box region shown in inset optical image of **a**. Inhomogeneous S$_Q$ distribution over the heterostructure region. The S$_Q$ intensity is dominant at the heterostructure region as indicated $I_Q > I_K$ (arrow) in **b**. **c** Extracted PL spectra, corresponding to A, B, and C positions: positive ($I_Q−I_K$) in A, which primarily covers the heterostructure region, $I_Q−I_K ≈ 0$ in B, and negative ($I_Q−I_K$) in C. **d** Schematic demonstration of the Q-band renormalization depending on the strong, medium, and weak coupling at the interface. **e, f** Experimental (top) and theoretical (down) photoluminescence for

oppositely-stacked WSe$_2$-WS$_2$: A compressive strain of respectively 0.3 and 0.15% in the top layer leads to peaks corresponding to S$_Q$ and Se$_Q$ in the experimental data. **g** Power-dependent PL spectra at high $I_Q$ intensities with strong coupling region. **h** In strong coupling regime, $\sigma$ changes the polarity from positive to negative with increasing the laser power and acts as an optical power switch (red dots). This crossing point of K and Q- exciton intensity with laser power is clearly visible in the inset contour plot. In medium coupling regime, $\sigma$ does not alter the polarity, although the power law is similar to that of strong coupling regime (green dots).

inverted with laser power (red dot, Fig. 4h). For example, $\sigma$ is positive in low power regime, whereas $\sigma$ is negative in high power regime. This distinct polarity flipping can be utilized as an optical power switch for monitoring solar panels. However, such a distinct polarity cannot be realized in an intermediate or weak coupling regime, as shown in the bottom panel of Fig. 4h, although the power law is similar to that in strong coupling.

## Discussion

In summary, we have observed additional PL peaks in TMD heterostructures that strongly rely on the stacking sequence of TMD monolayers. We ascribe these peaks to momentum-dark K-Q excitons, that follow a similar power law as typical K-K excitons. Numerous scenarios have been explored to explain such an unexpected stacking sequence dependence, including different substrates, different TMD materials, homogeneous bilayers, different number of layers, doping concentration, symmetric encapsulation of hBN or $SiO_2$, optical pathway, and low temperature. However, all these scenarios are not feasible to explain such unusual stacking sequence-dependent properties. On the other hand, our microscopic many-particle theory suggests layer-dependent strain as one possible solution, where assuming the top layer becomes more strained than the bottom layer due to actual stacking conditions. In reality, the strain landscape might be even more complicated as bubbles are known to appear in the heterostructure region. Thus, further studies are needed to fully understand the observed stacking sequence dependence. However, experimentally observed dark excitons of TMD heterostructures with inevitable use of substrate can be engineered for further exciton dynamics and can be useful as an optical power switch. In addition, intralayer indirect excitons in heterostructures may hold significant potential for future valleytronic applications.

## Methods

### Fabrication of vdW heterostructures

Various TMDs including, $WS_2$, $WSe_2$, $MoS_2$, and $MoSe_2$ were mechanically exfoliated for heterostructure stacking. $SiO_2$, quartz, $HfO_2$ films, and h-BN layers were used as the substrate. The presence of monolayer TMDs was confirmed by PL, Raman, and AFM. The aligned dry transfer technique was adopted for stacking of the selected flake on another designated flake. A water-soluble layer, polyvinyl alcohol (PVA), followed by polymethyl methacrylate (PMMA) was spin-coated on the substrate with two steps of 5 s at 500 rpm and subsequently 60 s at 3000 rpm. The coated substrate was then baked over the glass transition temperature. Monolayer TMDs were exfoliated on PVA/PMMA-coated substrate. The whole substrate was submerged in warm water such that PVA was dissolved and consequently, TMD/PMMA film was scooped by a customized holder with a 2-3 mm size hole (see Supplementary Fig. 1). The holder was then annealed for 12 min at 120 °C. This holder was loaded on the customized dry-transfer machine. Second, the bottom TMD was exfoliated directly on substrate, which was further aligned with the top TMD/PMMA. The temperature was maintained at 110 °C during stacking to remove residuals including contaminants and bubbles. Finally, the top PMMA was removed by rinsing in acetone-IPA-ethanol solutions sequentially.

### Atomic force microscopy (AFM)

AFM was performed in a Hitachi AFM5000II probe station using NSG30_SS tips in a tapping mode. Gwyddion software was used for extracting the flake height by the thickness difference between substrate and desired flake.

### Photoluminescence (PL) and Raman spectroscopy

PL and Raman were carried out in a commercial Ntegra Spectra II confocal system under room temperature with 532 nm laser excitation.

The laser beam was focused onto the sample by using an X100 objective lens (with a numerical aperture of 0.7).

### Device fabrication and STS measurements

For STS measurement we fabricated our heterostructure on an insulating layer (300 nm $SiO_2$) to avoid the band renormalization in TMDs due to the metal substrate. The contact electrode was constructed for grounding our sample. STS measurements were performed in an ultra-high vacuum (UHV) chamber with a base pressure of around $2.0 \times 10^{-11}$ Torr by using first-generation low-temperature Scienta Omicron (Germany) scanning tunneling microscopy (LT-STM) at room temperature. Electrochemically etched tungsten (W) tips were used for the measurement. Surface oxides of the tips had been removed by electron bombardment inside the UHV chamber. For taking STS data we used the conventional lock-in technique with a voltage modulation of 16 $mV_{rms}$ at a frequency of 817 Hz.

### Mechanical cleaning

The fabrication process often introduces residues, which reduces carrier mobility. Especially post-lithography cleaning is therefore required. By using contact mode AFM, the tip pressure allows pushing the contamination at the edges of the scan area, so that mechanical cleaning can be possible (Supplementary Fig. 25). This significantly improves the electronic properties and is much more beneficial for high vacuum measurement, as suggested previously[32].

### Electronic bandgap determination

Band onset is determined by dI/dV spectrum. The bandgap of the material, the energy difference by conduction band minimum (CBM) to valance band maximum (VBM) in dI/dV spectrum, was extracted by taking a logarithm of dI/dV spectrum[33].

### Theoretical modeling

Although the hybridization of dark states has been shown to have a significant impact on their energetic position and associated photoluminescence[34,35], the observed stacking-sequence-dependent peaks $S_Q$ and $Se_Q$ require to go beyond stacking-symmetric interlayer coupling effects. For this purpose, we consider a stacking-induced asymmetric strain and consider how this would change the monolayer PL[34,35]. We microscopically model the monolayer PL including indirect recombination of momentum-dark excitons via the recently introduced Elliott formula[31] $I(E) = \frac{2|M|^2[I_d + I_{ind}(E)]}{(E_{0,K-K} - E)^2 + (\gamma + \Gamma_{0,K-K})^2}$ with a state-independent inhomogeneous broadening $\Gamma_{inh}$ reflecting the experimental linewidth. Here $I_d = \gamma N_{0,K-K}$ provides the direct radiative recombination PL with rate $\gamma$ while $I_{ind}(E, t) = \sum_{\alpha,\beta,\pm} c_{\alpha;\beta}^{\pm} N_\alpha(t) \frac{\Gamma_\alpha + \Gamma_{inh}}{(E_\alpha \pm \epsilon_\beta - E)^2 + (\Gamma_\alpha + \Gamma_{inh})^2}$ provides the indirect recombination of excitonic states $\alpha = Q, \nu$ with energy $E_\alpha$ (having center-of-mass momentum Q in valley $\nu$). The indirect emission is assisted by absorption/emission of phononic modes $\beta$ with energy $\epsilon_\beta$ (via coefficients c depending on exciton-phonon interaction[31]). Finally, $\Gamma_\alpha$ and $N_\alpha$ provide respectively the excitonic scattering rate and thermalized occupation, both depending crucially on the exciton band alignment. The latter is evaluated by solving the Wannier equation after introducing a generalized Keldysh potential for the Coulomb interaction, where we include the dielectric constants $\epsilon$ for the environment ($\epsilon = 3.8$ and 1 for respective $SiO_2$ and air) and the pristine/strained electronic dispersion obtained from DFT studies[36]. In this way we microscopically predict in W-based materials a strain-induced increase of the energy separation between bright K-K and momentum-dark K-Q excitons, with a consequent increase in the corresponding occupations. This in turn results in an increased indirect emission from monolayer K-Q states in comparison to the direct one, as shown in Fig. 4e, f. In particular, the strain in the top layer induces new peaks in good agreement with the experimental observation. The phonon-assisted

emission has a lower efficiency than direct recombination, which is, however, compensated by a higher occupation of the K-Q excitons compared to the bright excitons. The higher relative occupation of K-Q excitons is induced by the energy alignment in tungsten-based materials and is further triggered by compressive strain. Furthermore, at decreased temperatures (77 K), we find an increased relative occupation of K-Q excitons and hence more intense phonon-sidebands[31]. In contrast, in Mo-based materials, K-Q excitons are higher in energy than the bright-excitons even in the presence of compressive strain, Supplementary Fig. 31. This leads to a negligible phonon-assisted PL. Nevertheless, the strain-induced reduction of the dark-bright exciton separation (Supplementary Fig. 32) could potentially lead to a resonance similar to $Se_Q$ in $MoSe_2/WS_2$, if additional activation mechanisms are present, e.g., via the interplay of strain and defects[37]. The microscopic evaluation of these mechanisms goes beyond the scope of this work.

## Data availability

The necessary source data needed to evaluate the key findings are provided in this paper. Further additional data that support the findings are available from the corresponding author upon request. Source data are provided with this paper.

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

## Acknowledgements

We thank Phillip Kim, Ivan Savenko, and Anirban Kundu for fruitful discussions. We thank Deok Soo Kim and Bumsub Song for the useful discussion related to time-resolved photoluminescence and scanning tunneling spectroscopy measurements respectively. This work was supported by the Institute for Basis Science of Korea (IBS-R011-D1) and Advanced Facility Center for Quantum Technology. The Marburg group acknowledges support from Deutsche Forschungsgemeinschaft (DFG) via SFB 1083 (Project B9) and the European Unions Horizon 2020 research and innovation program under grant agreement No 881603 (Graphene Flagship). Second harmonic generation measurements were

supported by the Center for Nanophase Materials Sciences (CNMS), U.S. Department of Energy, Office of Science User Facility at Oak Ridge National Laboratory.

## Author contributions

The idea was conceived by R.S. and supervised by Y.H.L. R.S. fabricated all vdW heterostructures and characterized them via atomic force microscopy, Raman spectroscopy, photoluminescence, and scanning tunneling spectroscopy. R.S. analyzed all data with the help of Y.H.L. R.S. fabricated the scanning tunneling devices with discussion with C.B. S.J.Y and A.P. measured stacking angle via second harmonic generation. K.P.D with R.S. characterized absorption measurement and R.S. analyzed the data. R.R., S.B., and E.M. provided theoretical support. R.S. and Y.H.L. wrote the manuscript. All authors discussed the results and review the manuscript.

## Competing interests

The authors declare no competing interests.
