## [Peer Review File · Nature Communications]

Reviewers' Comments:

Reviewer #1:

Remarks to the Author:

The manuscript by Riya Sebait et al. demonstrated a dark-exciton in TMDs vertical heterostructures fabricated by micromechanical exfoliation and manual restacking method. The authors found that the observed new PL emission was strongly related to the stacking sequence and the peak energy was slightly lower than the A-exciton of top materials. And the authors claimed that the new peak was attributed to the dark K-Q exciton of the top materials. Furthermore, the PL properties of a variety of TMDs vertical hetero- and homo-structures with different stacking sequences, different components, different substrates, and different thicknesses were investigated. The PL spectra taken from all these heterostructures showed the inherent existence of the dark K-Q excitons. Then, power dependent PL characteristic was studied to further elucidate the underlying mechanism of the dark K-Q exciton. The authors found that unlike exciton and trion, the peak energy of the dark K-Q exciton shows a downshift with the power of the excitation laser increase, which was attributed to strong charge screening. Finally, the authors found that the intensity of the dark K-Q exciton depends on the coupling at the interface. Overall, this manuscript is well organized and written. Nonetheless, before publication the following points should be properly addressed:

1. As the authors shown in this manuscript, the luminous intensity of dark K-Q exciton is relatively strong and can be compared with the A exciton, especially in top-WS₂/bottom-TMDs heterolayer (as shown in Figure 1). Hence, one of my main concerns is why the dark K-Q exciton hasn't been observed in previous references (such as Small 2019, 15, 1902424; Nat. Phys. 2018, 14, 801; Adv.Sci.2019, 6, 1802092; Nature 2022, 610, 478; Phys. Status Solidi B 2019, 256, 1900308.), after all, there are a lot of papers studying the optical properties of 2D TMDs vertical heterostructures prepared by restacking the mechanical exfoliation nanoflakes.
2. Did the authors find any interlayer excitons in these heterostructures? If yes, please add the corresponding results, if not, please give some explanations.
3. As shown in Figure 1f, the energy of the SeQ peak from top-MoSe₂/bottom-WS₂ heterostructure is 1.78 eV and larger than the A-exciton of MoSe₂, which differs from other heterostructures. The authors are advised to explain this anomaly.
4. The top-WS₂/bottom-MoS₂ and top-MoS₂/bottom-WS₂ heterostructures have not been prepared and studied. The authors are advised to add these results.
5. The authors were asked to give more analysis and explanation about the PL emission strength difference between A- and B-exciton of the MoS₂ from top-WSe₂/bottom-MoS₂ and top-MoS₂/bottom-WSe₂ heterostructures in Figure S10 a.
6. Figure 4b gives the PL peak position mapping of top-WS₂/bottom-WSe₂ heterostructure, please add the corresponding PL intensity mapping.
7. As shown in Figure 1, the PL emission strength of WS₂ collected from WS₂/TMDs and TMDs/WS₂ heterostructures on SiO₂/Si substrate is much stronger than that of other TMDs. However, as shown in Extended Data Fig. 3 and Extended Data Fig. 5, these of WS₂/TMDs and TMDs/WS₂ heterostructures on quartz and HfO₂ substrates show opposite results. Please give explanation.
8. The authors are advised to add the absorption and time-resolved PL characteristics. In addition, first-principles calculations of band structures of heterostructures could be useful for explaining the origin of the new PL emission.

Reviewer #2:

Remarks to the Author:

In the manuscript entitled "Sequential order dependent dark-exciton modulation in bi-layered TMD heterostructure", Sebait et al studied how stacking sequence affects momentum-dark exciton emission in TMDs heterobilayers. Since most of previous studies focus on understanding the optical properties of interlayer exciton states, it is quite interesting to see a detailed study on exploring intralayer exciton states. Although the authors present some experimental evidence combined with theoretical modeling to support their conclusions, I still have a few concerns on how they interpret their results. The authors need to fully address my following comments before I can recommend to publish on Nature Communications.

1. In the Fig. 1a, the authors observe a new emission state at 1.89 eV in the WS₂(top)/WSe₂(bottom) heterobilayer which they attribute to the indirect exciton (K-Q) emission. However, a few groups previously have fabricated similar samples and they don't see such features (e.g., ACS Nano 2016, 10, 6612–6622, Small 2019, 15, 1902424). How the authors explain the discrepancy?

2. In the Fig. 2c, the authors claim the momentum dark K-Q exciton states (SQ) lie ~ 100 meV lower than K-K excitons. According to the Boltzmann equation $(N(\text{bright exciton}))/N(\text{dark exciton}) = e^{(-\Delta E/k_B T)}$, ΔE is the energy difference between K-K and K-Q excitons, most of photo-excited excitons will populate in the energetically favorable K-Q states even at room temperature. So why direct exciton (K-K) recombination still contributes significantly in the PL emission as shown in the Fig. 1a and Extended Fig. 2?

3. In the Extended Fig. 2, the authors measure a few samples with different twist angles. It is interesting to see the emission intensity and energy of SQ change with twist angle. A previous study has shown the twist angle could significantly affect intralayer exciton states in WS₂-WSe₂ heterobilayer, specifically, the energy difference between K-Q and K-K excitons (Nature Materials. 2020, 19, 617–623). The authors need to carry out more careful analysis on the twist-angle-dependent behavior as it provides another solid evidence for their interpretations.

4. Finally, the authors theoretically explain the possible origin of K-Q dark exciton emission is due to top layer is more strained than the bottom layer. However, in the stacked heterobilayers, the formation of nanobubbles is unavoidable which could induce localized strain regions with exciton localization in both WS₂ and WSe₂ layers (Nature Nanotechnology, 2020, 15, 854–860). Moreover, as shown in the PL mapping (Fig. 4b), it could explain the inhomogeneous distribution of this new emission state. How the authors rule out this possible cause?

Reviewer #3:

Remarks to the Author:

In their manuscript entitled "Sequential order dependent dark-exciton modulation in bi-layered TMD heterostructures", the authors claim that the optical properties of TMD hetero- and homo-structures depends in the stacking order of the individual layers. It results in the emergence of a peak in the photoluminescence at a lower energy than the A-exciton of the top layer that the authors assigned to be an intralayer indirect exciton. The authors have performed a thorough set of experiments on various samples. In this sense, the observation of stacking dependent optical properties is convincing. The manuscript is well written and referenced, and a lot of additional data support their observation. However, given that the interpretation of the effect is mainly attributed to strain, which is known to induce direct to indirect transition of the optical bandgap in TMDs, the novelty of this work seems limited. Their result could potentially be worth publishing in Nature Communication provided that the authors clarify the novelty of their finding as well as a few points of concerns:

1) Previous works have demonstrated that strain can induce drastic change of the optical bandgap and thus PL response e.g. [Nano Lett. 13, 3626-3630 (2013)] and can be used to engineer the optical properties of TMDs. In this sense, the work presented by the authors seems like a direct application of this concept. Can the author clarify the novelty of their work with respect to prior art on strain-based bandgap engineering?

2) The authors do not show, or I missed it, the PL from interlayer excitons (ILX) in addition to their S_Q and S_{eQ} signals. I understand this is not the scope of this work, but that would remove any ambiguity on the possibility that the observed peaks are originating from an interlayer excitonic state. In particular, the authors claim to have tested a large number of samples that are randomly stacked. One would expect to see data exhibiting both the additional peak they observe and the signature (or not) of an ILX depending on twist angle. How random is the stacking? Have the authors measured the twist angle of their samples? Since the samples are heated up during the fabrication process this could favor relaxation of the top layer along some preferential orientations.

3) I am not sure I fully understand how that the authors can claim that the peak from the top layer originates from K-Q excitons. I understand that in TMD HS, K valleys of the two layers do not couple and that coupling between bands of the layers varies to be the strongest at Gamma. Is that why the Q-band is renormalized and not K? Is there a possibility for the indirect exciton to be Gamma-Q or Gamma-K (with holes at Gamma)? If this renormalization is more pronounced for strongly coupled layers, as the authors show, why doesn't it affect the Q band of the bottom layer? The authors perform STS measurements from which they conclude that the low energy band is located in the top layer. If there is a strong coupling between layers, shouldn't one expect an overlap of the wavefunctions between the layers? I am not sure I understand how they come to this conclusion and how they can separate contributions from the two layers.

4) If the change in the Q-band is related to strain, I am surprised not to see some similar signature in the bottom layer. I would be surprised that all the bottom layers of the studied samples are strain free. This brings me to the PL response seen in Fig. 1d, S8c, S15a, in which it is the clearest. In those measurements, the bottom layer shows a broader emission around the A-exciton than when it is on top. Could it be that the indirect exciton peak is also present for the bottom layer but with a much smaller energy separation from the A-exciton and a different relative intensity?

5) As the authors claim, emission from indirect excitons has to be phonon-assisted. That process should be very sensitive to temperature. Have the authors performed a temperature dependence to see whether the photoluminescence is in agreement with previous work, e.g. [Brem et al., Nano Lett. 20, 2849-2856 (2020)]?

6) In their final discussions, the authors suggest to use their finding for optical switch, which is an interesting approach. Have the authors considered other potential applications, such as valleytronics. Would it be possible to valley-polarize those indirect excitons, and take advantage of their long lifetime (with respect to the direct exciton)?

Responses to the reviewer's comments:

Reviewer: 1

General comment:

The manuscript by Riya Sebait et al. demonstrated a dark-excitons in TMDs vertical heterostructures fabricated by micromechanical exfoliation and manual restacking method. The authors found that the observed new PL emission was strongly related to the stacking sequence and the peak energy was slightly low than the A-exciton of top materials. And the authors claimed that the new peak was attributed to the dark K-Q exciton of the top materials. Furthermore, the PL properties of a variety of TMDs vertical hetero- and homo-structures with different stacking sequences, different components, different substrates, and different thicknesses were investigated. The PL spectra taken from all these heterostructures showed the inherent existence of the dark K-Q excitons. Then, power dependent PL characteristic was studied for further elucidate the underlying mechanism of the dark K-Q exciton. The authors found that unlike exciton and trion, the peak energy of the dark K-Q exciton shows a downshift with the power of the excitation lase increase, which was attributed to strong charge screening. Finally, the authors found that the intensity of the dark K-Q exciton depend on the coupling at the interface. Overall, this manuscript is well organized and written.

Response: We thank the reviewer for the careful and critical assessment of the manuscript and constructive comments. We appreciate all the valuable comments and suggestions, which eventually helped us to improve the manuscript's quality. We believe that the revised manuscript substantially improved its quality and readability. We have answered the queries raised by the reviewer in a pointwise manner in the following sections.

Comment 1: As the authors shown in this manuscript, the luminous intensity of dark K-Q exciton is relatively strong and can be compared with the A exciton, especially in top-WS₂/bottom-TMDs heterolayer (as shown in Figure 1). Hence, one of my main concerns is why the dark K-Q exciton hasn't observed in previous references (such as Small 2019, 15, 1902424; Nat. Phys. 2018, 14, 801; Adv.Sci.2019, 6, 1802092; Nature 2022, 610, 478; Phys. Status Solidi B 2019, 256, 1900308.), after all, there are a lot of papers studying the optical properties of 2D TMDs vertical heterostructures prepared by restacking the mechanical exfoliation nanoflakes.

Response: We thank the reviewer for raising the concerns regarding the previous studies where no such K-Q dark exciton is observed. We are also thankful to the reviewer for giving us the opportunity for clarifying the concerns. To the best of our knowledge, the fabrication process of the heterobilayer is one of the key factors for observing such K-Q dark excitons, because the strong interlayer coupling is deemed necessary for observing such phenomena. Furthermore, K-Q excitons are also dependent on the materials of the heterostructure. We will discuss here the previous reports that do show K-Q excitons unintentionally for given strong interlayer coupling, whereas such K-Q excitons are not observed in the case of weak interlayer coupling. In our case, we paid attention to having strong interlayer coupling during heterostructure

fabrication with different materials, during transfer without involving residues. Therefore, firstly we discussed the fabrication process and stacking sequence details and then compared with the reported results one by one.

(A) Strongly coupled clean interface:

We used an aligned dry-transfer technique for stacking the selected flake on another designated flake. The top layer was first exfoliated on PMMA/PVA-coated substrate. A customized holder with a 2-3 mm hole was used to scoop the TMD/PMMA film after dissolving the PVA in warm water. The bottom layer was directly exfoliated onto the substrate and aligned with the top TMD/PMMA and stacked together. The transfer temperature of 110 °C was maintained during staking to avoid interfacial residues. The stepwise stacking procedure was schematically shown in Supplementary Fig. S1 and discussed in detail in the method section (page 15). Figure R1-1a shows the K-Q exciton from the clean interface sample. Meanwhile, in another heterostructure, where the bottom layer was transferred with polymer (such as PMMA) to the target substrate, and then cleaned the polymer by using the conventional acetone-IPA-ethanol cleaning process, followed by the top layer was transferred on top of the bottom layer. Even though polymers were cleaned, some unwanted contaminants still remained, and a clean surface can be hardly achieved. As a consequence, we did not find any clear signature of Q-band-related dark exciton peak (S_Q) in this case (Fig. R1-1b). Meanwhile, the cleanliness-dependent strong coupling has been confirmed by the presence of A_{21g} Raman mode of WSe_2 (Supplementary Fig. S5) exclusively at the clean heterobilayer region. A homogeneous topographic image by AFM, from the clean heterobilayer region was also confirmed in Supplementary Fig. S6. The detail has been described in supplementary information note 3 (SI page 5).

Fig. R1-1 (Supplementary Fig. S4) **Interface cleanliness.** **a**, Schematic representation of WS_2/WSe_2 heterostructure; A clear S_Q peak was observed from the clean interface by our transfer method. **b**, Schematic representation of contaminated heterostructure; At the contaminated interface, the S_Q peak was invisible.

Considering the reviewer's comments, we added the above schematic in Supplementary Fig. S4.

(B) Comparison with previous reports:

(i) **Small 2019, 15, 1902424:**

Stacking sequential order: WSe₂ (top layer)/WS₂ (bottom layer)

Fabrication method: The WS₂ was directly exfoliated on a substrate (SiO₂) and WSe₂ was exfoliated on polymethyl-methacrylate (PMMA) and followed by transferring the WSe₂ layer on top of the WS₂ layer via dry transfer technique. Subsequently, the as-prepared WSe₂/WS₂ hetero-bilayers (HBs) were annealed in an Ar atmosphere (3 Torr, 200 Scm) at 300 °C for 12 hours to enhance the vdW interaction between the layers. The procedure is considered a clean interface.

Comparison of the results: Authors observed interlayer exciton around 1.5 eV at low temperatures (<100K) (Fig. R1-2a) and close to 1.55 eV at room temperature. This interlayer exciton energy difference from inlayer WSe₂ exciton was around 70-80 meV (shown in green dotted lines in Fig. R1-2a inset), different from other previous reports [Nature 2019, 567, 76; Optical Express 2020, 28, 13260; Nature Materials 2020, 19, 617]. Such an energy difference of interlayer exciton matches well with our K-Q exciton (Se_Q peak in Fig. R1-2b). We further like to emphasize that interlayer exciton strongly depends on angle alignment between the layers as described previously [Nature 2019, 567, 76; Nature 2022, 610, 478; Nature Materials 2020, 19, 617; Nature Communications 2020, 11, 5888; ACS Nano 2017, 11, 4041]. However, in their work authors did not mention the angle alignment between the layers. In other words, the observation of interlayer exciton in their case is rather ambiguous.

Figure R1-2 | Comparison with previous work. **a**, Top: Schematic and optical image of WSe₂/WS₂ heterobilayer. Bottom: PL spectra at WSe₂/WS₂ heterobilayer region, an additional peak emerges at the heterostructure region called an interlayer exciton (X_{IL}) [Reference: Small 2019, 15, 1902424]. **b**, Our results for the similar stacking configuration to **a**, an additional peak at heterostructure (cyan color), which is called Se_Q peak emerging due to the K-Q intralayer excitons.

(ii) *Nat. Phys.* **2018**, *14*, 801:

Stacking sequential order: WSe₂ (top layer)/MoS₂ (bottom layer)

Fabrication method: Both MoS₂ and WSe₂ exfoliated onto polydimethylsiloxane (PDMS) substrates. First MoS₂ flake was transferred onto the target substrate. Subsequently, the WSe₂ flake was transferred on top of MoS₂. Afterward, annealing the heterostructure at 150° C for 5 hrs in an H₂/Ar gaseous environment. This annealing process promotes strong interfacial coupling, resulting in the clean interface.

Comparison of the results: This article also reported an additional exciton peak only at the heterostructure region at the energy of 1.6 eV, 60-70 meV lower than WSe₂ neutral exciton (Fig. R1-2a). Authors claim that this additional peak appeared due to interlayer exciton between WSe₂-MoS₂ heterobilayer. In this case, interlayer exciton always appeared independent of the stacking angle, in contrast to other previous reports [Nature 2019, 567, 76; Nature 2022, 610, 478; Nature Materials 2020, 19, 617; Nature Communications 2020, 11, 5888; ACS Nano 2017, 11, 4041]. Further, the intensity at room temperature was comparable to or higher than that of neutral exciton of each layer, again distinct from previous reports [Phys. Status Solidi B 2019, 256, 1900308; Nature 2019, 567, 76; Nature 2022, 610, 478; Small 2019, 15, 1902424; Nature Materials 2020, 19, 617; Optics Express 2020, 28, 13260]. This cannot be explained by non-equilibrium effects alone. The authors justified the reason by considering Γ -K indirect transition instead of K-K interlayer transition.

This additional peak energy below the inlayer WSe₂ peak is also similar to that of the K-Q exciton (Se_Q peak) (Fig. R1-2b) in our case. However, experimentally we observed that the additional peak (Se_Q peak) vanishes for altering the stacking sequential order (Supplementary Fig. S12a), even though other parameters remain identical. Furthermore, the authors mentioned that the additional PL peak explained by interlayer exciton is independent of the stacking angle, which is again similar to our finding. Authors claim that this additional peak is due to Γ -K interlayer exciton, which is different from our claim.

Figure R1-3 | Comparison with previous work. a, PL spectra at WSe₂/MoS₂ heterobilayer region, an additional peak emerges below the inlayer WSe₂ exciton [Reference: *Nat. Phys.* **2018**, *14*, 801]. **b,** Our results, similar to **a**, also had the same additional peak at heterostructure, which we called Se_Q peak emerging due to the K-Q intralayer excitons.

(iii) *Adv.Sci.*2019, 6, 1802092:

Stacking sequential order: MoS₂ (top layer)/ WS₂ (bottom layer)

Fabrication method: Mechanically exfoliated WS₂ flakes were transferred to the viscoelastic PDMS, after that PDMS film was attached to the target substrate. Using the dry transfer method, the top MoS₂ flakes were transferred to the target WS₂ layer.

Comparison of the results: Authors demonstrated the strong coupling between interlayer excitons in MoS₂/WS₂ heterostructure via cavity-enhanced Mie resonances in silicon nanoparticles. Authors did not observe any interlayer exciton peak on commercially available SiO₂/Si substrate, whereas the interlayer exciton was only available from silicon nanoparticles on SiO₂/Si substrate. Further, they demonstrated that coherence stacking gives a prominent interlayer exciton peak compared to random stack. Their approach is to observe interlayer exciton on Si nanoparticles, which is not relevant for our research target to observe K-Q exciton peaks on stacking sequential order.

(iv) *Nature* 2022, 610, 478:

Stacking sequential order: WS₂ (top layer)/ WSe₂ (bottom layer) (main manuscript) and WSe₂ (top layer)/ WS₂ (bottom layer) (Supplementary).

Fabrication method: Bottom WSe₂ layer was dry-transferred from the gel film to the substrate first. The second layer of WS₂ was then transferred with a thermal heating process at 363 K. Heating the layers improves the interlayer coupling between them. Therefore, this procedure is considered a clean interface.

Comparison of the results: Authors observed several emissions due to moiré-trapped interlayer excitons in suspended WSe₂/WS₂ heterobilayer. Reduced screening in free-standing heterobilayer, led to more robust dipole-dipole interactions, that confirm the visibility of interlayer excitons. They observed the nature of the dipole interactions among free interlayer excitons; from repulsive to attractive, caused by quantum-exchange-correlation effects, which lead to the appearance of interlayer biexciton phases.

Further, the authors demonstrated the PL spectra on both WSe₂/WS₂ and WS₂/WSe₂ heterobilayer in Fig. R1-4 (see also their *Extended Data Fig. 3b and c*). As shown in Fig. R1-4a, they observed some additional PL peaks (red box area) below the inlayer WS₂ peak (similar to our S_Q peak), whereas such peak was not observed in the reverse stacking sequence of WSe₂/WS₂ heterostructure (Fig. R1-4b). On the other hand, in Fig. R1-4b, there was another additional peak (similar to our Se_Q peak) below WSe₂ inlayer exciton (marked in red circle), again absent in the alternative stacking case as shown in Fig. R1-4a. Nevertheless, the authors did not pay attention to these results and never explained origin of such peaks, which could be related to K-Q excitons in our study.

Figure R1-4 | Comparison with previous work. a,b, PL spectra of WS_2/WSe_2 and WSe_2/WS_2 heterobilayer, an additional peak emerges below the top layer exciton similar to our S_0 and Se_0 peak respectively [Reference: Nature 2022, 610, 478]. The red box and circle have been drawn to represent the peak position, which is present or absent depending on stacking sequential order.

(v) Phys. Status Solidi B 2019, 256, 1900308:

Stacking sequential order: MoS_2-WSe_2 and $MoSe_2-WSe_2$. Any specific stacking sequential order has not been mentioned.

Fabrication method: TMD flakes were exfoliated onto PDMS substrate from bulk crystal, subsequently transferring the constituent flakes of a heterostructure onto the target substrate. MoS_2-WSe_2 heterostructure was randomly oriented to each other, whereas crystallographic angles were aligned in $MoSe_2-WSe_2$ heterostructure. This is considered a strong coupling.

Comparison of the results: Authors presented the experimental and theoretical studies of interlayer exciton in two different heterobilayer combinations such as twisted MoS_2-WSe_2 and aligned $MoSe_2-WSe_2$. In both cases, interlayer excitons have been observed. However, these interlayer excitons show different properties. In the MoS_2-WSe_2 heterostructure, interlayer exciton was observed near 1.6 eV, 60-70 meV below the WSe_2 inlayer exciton, and this interlayer exciton is independent of the stacking angle between the layers (Fig R1-5a). Further, authors claim from theoretical calculations that this interlayer exciton appears due to indirect $\Gamma-K$ transition instead of $K-K$ interlayer transition. On the other hand, in the $MoSe_2-WSe_2$ heterostructure, they got interlayer exciton of more than 200 meV separated from residual monolayer emission (Fig R1-5b). In this case, authors claim that it is a $K-K$ transition and strongly depends on the twisted angle between the layers. Although authors did not clearly indicate the stacking sequential order, which is an important parameter to compare with our result. We presume that the WSe_2 layer is located on the top layer of the MoS_2 layer as per their previous report [Proc. Natl. Acad. Sci. 2014, 111, 6198].

Figure R1-5 | Comparison with previous work. a,b, PL spectra of WSe₂/MoS₂ and WSe₂/MoSe₂ heterobilayer, an additional peak emerges at the heterobilayer region called interlayer excitons (ILE) [Reference: *Phys. Status Solidi B* 2019, 256, 1900308]. c,d, Our results with similar stacking configuration to a,b.

In our approach, in the top WSe₂/bottom MoS₂ heterostructure (Fig R1-5c), we observe a new additional PL peak (K-Q peak) below the inlayer WSe₂ peak. We interpret our peak as K-Q intralayer exciton. If we reverse the stacking sequential order to the top MoS₂/bottom WSe₂ heterostructure, such K-Q exciton disappears, which is well distinguished from interlayer exciton in their work. In the case of MoSe₂-WSe₂ heterostructure, we have never observed any Se_Q peak due to overlapping the energies between two materials as we have mentioned in our manuscript (Supplementary Fig. S12b). However, it would be noted that in our case we have also observed a tiny signal of interlayer exciton around the energy 1.35 eV, similar to this mentioned report at room temperature (inset of Fig R1-5d).

(C) Comparison with previous reports (additional):

While the reviewer suggested a correlation between the previous references to K-Q excitons, we like to include a few more reports to provide PL evidence of K-Q exciton in TMD heterostructures depending on the stacking sequential order. Since Reviewer 2 also raised a similar question, we have included the mentioned references in this section as well.

(vi) *Nano Lett.* 2014, 14, 3185-3190:

Stacking sequential order: WS₂ (Top layer)/ MoS₂ (bottom layer) (main manuscript) and MoS₂ (top layer)/ WS₂ (bottom layer) (Supplementary).

Fabrication method PDMS was spin-coated on CVD-grown monolayer WS₂ and PDMS/WS₂ was then separated from SiO₂/Si substrate by KOH solution and further transferred to monolayer MoS₂ via PDMS stamping techniques. Further, the cleanliness of the interface was improved by annealing the heterostructures.

Comparison of the results: Authors demonstrated the tuning of interlayer coupling via thermal annealing. In the strongly coupled region, authors found two additional peaks along with inlayer excitonic peaks, named – P_{hetero} and P_{indirect} at energy 1.94 and 1.75 eV respectively (Fig. R1-6a). The P_{indirect} peak position changes rapidly with annealing time and is attributed to phonon-assisted indirect band gap transition. The P_{hetero} peak position is ~ 60 meV below the inlayer WS₂ peak, and this amount of energy shift is lower than the accuracy of DFT calculations. Further, authors discussed the P_{hetero} emission in terms of excitonic effects and ultrafast charge relaxation across the heterostructure. However, it does not explain the distinct behavior of MoS₂, leaving the origin of the P_{hetero} emission elusive.

In Supplementary data, on the other hand, authors also fabricated the reverse stacking (MoS₂(top)/WS₂(bottom)) and in this case, surprisingly the P_{hetero} peak vanishes (Fig. R1-6c). Authors did not pay attention to this as it is very unconventional. This report strongly supports our work. For clear demonstration, we also fabricated a similarly configured heterostructure and measured the PL. WS₂/MoS₂ heterostructure reveals the S_Q peak (which authors named P_{hetero} in their work), while MoS₂/WS₂ heterostructure vanishes the S_Q peak (Fig. R1-6 b,d). This clearly demonstrates that the origin of the K-Q exciton peak is determined by the stacking sequential order.

Figure R1-6 | Comparison with previous work. **a,b**, PL spectra of WS₂/MoS₂ heterostructure of previously reported [Reference: *Nano Lett.* 2014, 14, 3185-3190] and our result. **c,d**, a similar result for inverse stacking MoS₂/WS₂.

(vii) ACS Nano 2016, 10, 6612–6622:

Stacking sequential order: WSe₂ (top layer)/WS₂ (bottom layer) (main manuscript) and WS₂ (top layer)/WSe₂ (bottom layer) (Supplementary)

Fabrication method: Heterostructures were prepared by the PMMA stamping method. First, PMMA was coated on the CVD-grown WS₂ flake grown on a sapphire substrate. The WS₂ flake was detached from sapphire by KOH solution and thereafter transferred to ML WSe₂ flake at 100 °C. Heterostructures were then annealed at 350 °C to remove the polymer residues.

Comparison of the results: Authors never observed any clear interlayer exciton peak due to interfacial trap/residue between the layers (as shown in the AFM image in supplementary Fig. S4 and here Fig. R1-7a). The interlayer coupling has been improved via thermal annealing, which makes efficient charge separation and enhancement of light absorption at heterostructure. In WSe₂(top)/WS₂(bottom) heterostructure, a small additional PL peak (~830 nm) was

observed at the heterostructure (marked circle in Fig. R1-7b). This peak vanishes in the reverse stacking case (Fig. R1-7c). However, the new peaks were not that prominent due to remaining interfacial residues but still had a significant signature that matches well with our observation, however, authors did not pay attention to it.

Figure R1-7 | Comparison with previous work [Reference: ACS Nano 2016, 10, 6612–6622]. **a**, AFM mapping demonstrates the removal of interfacial contamination before and after annealing. **b,c**, PL spectra of WSe₂/WS₂ and inverse (WS₂/WSe₂) heterostructure.

(viii) Optics Express 2020, 28, 13260

Stacking sequential order: WSe₂ (top layer)/WS₂ (bottom layer).

Fabrication method: Each of the monolayer flakes has been exfoliated from the bulk crystal onto the silicon substrate. After that, the heterostructure was constructed by the fixed point transfer technique. However, the fabrication details have not been provided in this work.

Comparison of the results: Authors demonstrated the existence of double indirect interlayer exciton in WSe₂/WS₂ heterobilayer at the cryogenic temperature at energy 1.4585 eV and 1.4885 eV. At room temperature, authors observed a single interlayer exciton below the inlayer WSe₂ peak (Fig. R1-8a). Their interlayer exciton peaks are similar to that of our K-Q excitons (Se_Q peak in Fig. R1-8b), although our interpretation is very different from theirs. We emphasize that we confirmed Se_Q peaks from stacking sequential order.

Figure R1-8 | Comparison with previous work. a,b, PL spectra of WSe₂/WS₂ heterostructure of previously reported [Reference: Optics Express 2020, 28, 13260] and our result. The red dotted circle indicates the emergence of an additional peak at the heterostructure region.

(ix) Nature 2018, 560, 340:

Stacking sequential order: WSe₂ (top layer)/ MoS₂ (bottom layer)

Fabrication method: Flakes were first exfoliated on a polymer. The bottom MoS₂ layer was transferred to the h-BN substrate first. The second layer (WSe₂) was then transferred and encapsulated by another h-BN. Heterostructures were thermally annealed in a high vacuum. The fabrication procedure is presumably sound.

Comparison of the results: Authors demonstrated gate-modulated exciton flux in WSe₂/MoS₂ heterobilayer at room temperature. Type-II band alignment in WSe₂/MoS₂ results in effective charge separation between the constituent materials and forms indirect interlayer exciton, 75 meV below the intralayer WSe₂ monolayer exciton (Fig. R1-9). Further, authors claim that this interlayer exciton does not depend on the angle alignment between the layers due to the indirect nature of transition (in reciprocal space) and to the considerable lattice mismatch (about 4 %) between two layers. Further, we like to note that, at room temperature, in general, interlayer exciton intensity is low compared to intralayer [Phys. Status Solidi B 2019, 256, 1900308; Nature 2019, 567, 76; Nature 2022, 610, 478; Small 2019, 15, 1902424; Nature Materials 2020, 19, 617; Optics Express 2020, 28, 13260]. In this work, interlayer exciton intensity prevails compared to intralayer exciton at room temperature. This high interlayer exciton intensity makes it elusive as it is rare to observe such high intensity of indirect interlayer coupling at room temperature. Further, experimentally we observed that additional peak (Se_Q peak) vanishes for altering the stacking sequential order (Supplementary Fig. S12a), even though other parameters remain identical.

Figure R1-9 | Comparison with previous work [Reference: Nature 2018, 560, 340]. An additional PL peak at energy 1.57 eV in WSe₂/MoS₂ heterostructure.

In conclusion, the comparisons with previous reports are summarized below:

- (i) Optical anomalies of emergence or disappearance of exciton peaks have been observed recently from few reports that depend on stacking sequential order [Nature 2022, 610, 478; Nano Lett. 2014, 14, 3185-3190; ACS Nano 2016, 10, 6612–6622]. However, experimental verification of these anomalies is always ignored.
- (ii) Different combinations of TMDs generate distinct interlayer excitonic peaks, but previous reports have shown almost the same energy for different combinations, for example, WSe₂-WS₂ [Optics Express 2020, 28, 13260; Small 2019, 15, 1902424] and WSe₂-MoS₂ [Nature 2018, 560, 340; Phys. Status Solidi B 2019, 256, 1900308; Nat. Phys. 2018, 14, 801].
- (iii) Interlayer exciton should be independent of stacking sequential order [Nature 2022, 610, 478–484; Nat. Commun. 2015, 6, 4–9]. However, we have observed that by altering the stacking sequential order exclusively and keeping all the other parameters (experimental conditions) the same, our heterostructure peak emerges or vanishes depending on stacking sequential order. To date, there is no such report on this unusual phenomenon.

In our opinion, all these previous findings are actually supporting our results. For example - if WSe₂ is the top layer in a hetero-bilayer, we will then have a strong K-Q intralayer peak corresponding to WSe₂, independent of the bottom layer. All such inconsistencies arise from the absence of no such systematic study that extensively investigates a variety of different TMDs combinations.

We thank again to the reviewer for this query, which provided a more clear picture of our findings and we have added the above-mentioned references in the revised version of the manuscript. In addition, we are obliged to modify the introduction accordingly due to a few references mentioned by the reviewer (references: 9-13 in the revised manuscript).

Modifications to the manuscript (page 2, line 28):

In addition, they also show a variety of optically forbidden dark excitons, either due to spin-flip or momentum transfer.⁵⁻⁷ For example, both K-K bright and momentum-forbidden K-Q excitons emerge in monolayer WSe₂, as confirmed by time- and angle-resolved photoemission spectroscopy.⁶ These momentum-forbidden indirect excitons can also be realized in van der Waals (vdW) heterostructures, where bound electrons and holes are localized in two different layers, which is called an interlayer exciton.⁷⁻¹⁴ Emergence of these indirect interlayer excitons strongly depends on the stacking angle between the layers^{8,10,12} and is independent of stacking sequential order.^{10,15} Despite this, a few recent reports have accidentally observed optical anomalies in heterostructures that rely on the sequential order of the layers.^{15,16} However, the underlying mechanisms that give rise to these anomalies are not yet fully realized. This lack of understanding may be attributed in part to the hypothesis that altering the stacking sequence between layers would not significantly affect the intrinsic properties of heterostructures. Therefore, it is important to explore stacking sequential order dependence to further use of real device applications.

In this work, we tune dark excitons by changing the stacking sequence between TMD layers, whereas other experimental parameters remain constant. Such emergence or disappearance of dark excitons depending upon stacking sequence has not been reported to date.

Comment 2: Did the authors find any interlayer excitons in these heterostructures? If yes, please add the corresponding results, if not, please give some explanations.

Response: We are thankful to the reviewer for raising this important query. We have observed only a few times the signature of the interlayer exciton (ILE). For example, in MoSe₂/WSe₂ heterostructure, a weak ILE peak was observed at energy 1.35 eV at room temperature (Fig. R1-10), which is well matched with the previously reported results [Nature Communications 2015, 6, 6242; Phys. Status Solidi B 2019, 256, 1900308; Nature Communications 2021,12,1656]. However, in most of the cases, we have not observed any signature of interlayer excitons. The reason behind this is listed below -

1. In general, interlayer exciton requires specific angle alignment between the layer as observed previously [Nature 2019, 567, 76; Nature 2022, 610, 478; Nature Materials 2020, 19, 617; Nature Communications 2020, 11, 5888; ACS Nano 2017, 12, 4041]. However, in our case, heterostructures are not fabricated with angle alignment.

2. Interlayer excitons are most prominent at low temperatures [Phys. Status Solidi B 2019, 256, 1900308; Nature 2019, 567, 76; Nature 2022, 610, 478; Small 2019, 15, 1902424; Nature Materials 2020, 19, 617; Optics Express 2020, 28, 13260], nevertheless, in this work all the experiments have been conducted at room temperature unless specified.

3. Interlayer exciton generally appears, i.e., 1.35-1.45 eV as reported previously [Nature 2019, 567, 76; Nature 2022, 610, 478; Nature Materials 2020, 19, 617; Optics Express 2020, 28, 13260], which is beyond the scope of this work.

We believe that because of all these reasons stated above, the interlayer exciton does not appear in our case, except few of the samples, one of them shown below.

Figure R1-10 (Supplementary Fig. S10) | Interlayer exciton. An interlayer exciton (ILE) peak appears at energy 1.35 eV in MoSe₂/WSe₂ heterostructure at room temperature.

In the revised version of the supplementary, we have added a new figure (Supplementary Fig. S10) as well as a new paragraph to discriminate the interlayer exciton from our additional new PL peaks (S_Q/Se_Q).

Modifications to the manuscript (page 3, line 79):

Furthermore, the energies of these additional peaks (1.89 and 1.56 eV) are distinct from reported interlayer excitons (1.35-1.42 eV) for the WS₂-WSe₂ heterostructure.^{8,17} Moreover, we observed weak interlayer exciton peak at an energy of 1.35 eV in MoSe₂/WSe₂ heterostructure at room temperature (Supplementary Fig. 10). This is well matched with previously reported results,^{10,12,19} which is rarely observed as it is contingent upon a specific stacking angle.

Modifications to the Supplementary (page 10, line 154):

Note 6. Interlayer exciton

We rarely observed the interlayer exciton (ILE). For example, in MoSe₂/WSe₂ heterostructure, a weak ILE peak was observed at an energy of 1.35 eV at room temperature (Supplementary Fig. S10), which is well matched with previously reported results.⁴⁻⁶ However, in most of the cases, we have not observed any signature of interlayer excitons. The reason behind this is listed below:

i) In general, interlayer exciton requires specific angle alignment between the layers as observed previously.⁷⁻¹¹ However, in our case, heterostructures are not fabricated with specific angle alignment.

ii) Interlayer excitons are most prominent at low temperatures.^{5,7-9,12,13} Nevertheless, in this work, all the experiments have been conducted at room temperature.

iii) Interlayer exciton generally appears, i.e., at the energy of 1.35-1.45 eV for WSe₂-WS₂ heterostructure as reported previously,^{7-9,13} which is beyond the scope of this work.

Fig. S10 | Interlayer exciton. Interlayer exciton (ILE) peak appears at an energy of 1.35 eV in MoSe₂/WSe₂ heterostructure at room temperature.

Comment 3: As shown in Figure 1f, the energy of the SeQ peak from top-MoSe₂/bottom-WS₂ heterostructure is 1.78 eV and larger than the A-exciton of MoSe₂, which differ from other heterostructures. The authors are advised to explain this anomaly.

Response: The reviewer raises an important point, as the new peak Se_Q/S_Q for the top layer has a completely different energy position for W- and Mo-based materials, being respectively energetically below and above the bright exciton peak. This different behavior again indicates that K-Q excitons are responsible also for the peak Se_Q appearing in MoSe₂. To see this, in Fig. R1-11, we show the excitonic dispersion for K-Q excitons in (a) WS₂ and (b) MoSe₂ monolayers with and without strain (red and blue, respectively). The excitonic energies are microscopically determined by numerically solving the Wannier equation.

Comparing the energy of K-Q excitons to the K-K exciton energy E_{K-K} , we see that K-Q excitons are energetically below and above K-K (see the y-axis value negative and positive respectively) for (a) WS_2 and (b) $MoSe_2$, respectively - exactly as the peaks S_Q and Se_Q in Fig. 1e-f, respectively. In addition, compressive strain further increases the energy separation between K-Q and K-K excitons resulting in a higher relative occupation of dark excitons in comparison to bright excitons. Since an increased population could result in a stronger PL signal, these results indicate the compressive strain in the top layer as a possible activation mechanism for the Se_Q/S_Q peaks. In the case of $MoSe_2$ our theory predicts a slightly smaller dark-bright energy separation than expected from the experimental data in Fig. 1f. Here, we also predict a negligible phonon-assisted photoluminescence, suggesting additional activation mechanisms in the experiment. To give an example, in a recent study, the interplay of (tensile) strain and defects has been considered as a possible activation mechanism [Nature Commun. 13, 7691 (2022)].

Figure R1-11 (Supplementary Fig. S27): Dispersion of K-Q excitons with respect to the K-K exciton energy for **a**, WS_2 , and **b**, $MoSe_2$ monolayers with and without compressive biaxial strain. Note the different scaling on the y-axes (including negative and positive values).

In the revised version of the Supplementary, we have added a new figure (Supplementary Fig. S27) as well as a new paragraph including additional references to discuss how i) the different energy alignment of K-Q and K-K excitons in W- and Mo-based monolayers as well as ii) the redshift of their energy separation could be important indications of K-Q excitons being the origin of the experimentally observed S/S_{eQ} peaks.

Modifications to the manuscript (page 8, line 207):

By applying a small compressive strain on the top layer (top- WS_2 layer: 0.3% and top- WSe_2 layer: 0.15% strain in heterostructures), we can successfully reproduce the experimental data (Fig. 4e,f) for W-based materials. While the theoretical results shown in Fig. 4e-f are obtained for 77K for better visualization, a strain-induced increase of the energy separation between K-Q and K-K excitons (Supplementary Fig. 27 and Ref. [31]) could lead to visible phonon-sidebands also at higher temperatures, in accordance to experimental observations of Se_Q and S_Q . In contrast, Mo-based material showed K-Q dark exciton higher in energy compared to K-K bright exciton even in the presence of compressive strain (Details in Supplementary Fig 27). This leads to a negligible phonon-assisted PL. Therefore, the presence of a strained top layer, leads to a new peak in $WS_2/MoSe_2$, corresponding to lower energy K-Q states, while no new

phonon-assisted peaks are observed in MoSe₂/WS₂ due to higher energy K-Q excitons, resulting in negligible phonon-assisted PL (Supplementary Fig. 28).

Modifications to the Supplementary (page 25, line 406):

Note 21. Theoretical calculations for comparing the K-Q to K-K excitons for W- and Mo-based materials.

The excitonic energies are microscopically evaluated starting from the unstrained single-particle dispersion²⁵ and including their strain-dependent variations²⁶. These values are then used to numerically solve the Wannier equation by introducing a generalized Keldysh potential for the Coulomb interaction.

In Supplementary Fig. S27 top, we show the resulting excitonic center-of-mass dispersion of K-Q excitons for the WS₂ case, considering SiO₂/air as a dielectric environment. Already in the unstrained case (blue), the minimum K-Q energy is smaller than the bright-exciton energy E_{K-K} . In the presence of a compressive strain $s=-0.30\%$, the energy separation increases by a factor of 3 from 30 meV up to almost 87 meV. In contrast, in MoSe₂ the K-Q valley is energetically above the K-K valley as shown in Supplementary Fig. S27 bottom. In this case, compressive strain also leads to a blueshift of K-Q energies (redshift of E_{K-K}). This reduces the energy separation between the K-Q and K-K excitons from 123 to 69 meV, while keeping K-K as the ground state.

Fig. S27 | Dispersion of dark excitons. Energy of K-Q excitons as a function of the center-of-mass momentum for unstrained (blue) and compressively strained (red) (a) WS₂ and (b) MoSe₂.

In Supplementary Fig. S28, we provide a theory-experiment comparison for the WS₂-MoSe₂ heterostructure (Fig. 4e-f for WS₂-WSe₂). In the presence of a strained top layer, we find a new peak in WS₂/MoSe₂ in accordance with the experimentally measured S_Q , reflecting K-Q states with energy smaller than E_{K-K} (Supplementary Fig. S27 top). In contrast, we find no new phonon-assisted peaks in MoSe₂/WS₂, as here the K-Q excitons are energetically above the bright ones (Supplementary Fig. S27 bottom). This implies their reduced occupation in comparison to the bright-exciton states, resulting in negligible phonon-assisted

photoluminescence. Nevertheless, our prediction of a decreased $E_{K-Q}-E_{K-K}$ separation with compressive strain (Supplementary Fig. S27 bottom) could potentially lead to a peak similar to Se_Q in $MoSe_2/WS_2$, if an additional activation mechanism is present, e.g. via interplay of strain and defects.²⁷ The microscopic evaluation of these mechanisms goes beyond the scope of this work.

Fig. S28 | Experimental and theoretical PL for $MoSe_2$ - WS_2 heterostructure. **a**, In the case of top- WS_2 , a low-energy peak (S_Q) stems from the phonon-assisted transition by assuming a 0.45 % compressive strain in the WS_2 layer. **b**, In the case of the top- $MoSe_2$ layer, we do not see a phonon sideband on the high-energy side of the bright $MoSe_2$ exciton, as the occupation of the K-Q state is negligibly small.

Comment 4: The top- WS_2 /bottom- MoS_2 and top- MoS_2 /bottom- WS_2 heterostructures have not been prepared and studied. The authors are advised to add these results.

Response: We thank the reviewer's suggestion regarding the addition of WS_2 (top)/ MoS_2 (bottom) and inverted stacking MoS_2 (top)/ WS_2 (bottom) data. Considering this comment, we have fabricated and measured the PL of WS_2 - MoS_2 heterostructure for studying the stacking sequence-dependent properties. In WS_2 (top)/ MoS_2 (bottom) case, the notable S_Q peak has been observed as WS_2 as the top layer with an energy of 1.89 eV, similar to other TMDs combinations (red color peak in Fig. R1-12a). On the other hand, in MoS_2 (top)/ WS_2 (bottom) heterobilayer case we have not observed any dark exciton-related PL peak (Fig. R1-12b). This can be explained with MoS_2 being in the top layer where the K-Q peak for Mo-based material then emerges at higher energy than MoS_2 A-exciton (as we observed for the $MoSe_2$ case: Fig. 1f) and this K-Q energy overlaps with inlayer WS_2 exciton. Another reason could be the energy of the Q-band, which is located far away from the K-band [Phys. Rev. B 101, 201405 (2020)], subsequently incapable of band renormalization of K-Q

excitons. The B-exciton of MoS₂ is also indistinguishable for the same overlapping of the energy with inlayer WS₂ exciton.

We note that a similar PL peak to our S_Q peak has been observed previously in WS₂/MoS₂ heterobilayer [Nano Lett. 14, 3185-3190, 2014]. They do not have a clear picture of this particular peak and they claim it as P_{hetero}. Please see comment 1 for details.

Figure R1-12 (Supplementary Fig. S12) | Stacking sequence effect MoS₂-WS₂ combinations. a,b, PL spectra of MoS₂/WS₂ and inverted WS₂/MoS₂ heterostructure. S_Q still emerges on the WS₂ layer of the WS₂/MoS₂ heterostructure, whereas the S_Q peak in MoS₂ is indistinguishable.

We have added these data in the revised version of the supporting information (Supplementary Fig. S12). We have also added a new paragraph discussing their sequence-dependent properties.

Modifications to the Supplementary (page 13, line 194):

In WS₂(top)/MoS₂(bottom) case, the notable S_Q peak has been observed as WS₂ as the top layer with an energy of 1.89 eV, similar to other TMDs combinations (red color peak in Supplementary Fig. 12c, left). On the other hand, in MoS₂(top)/WS₂(bottom) heterobilayer case, we have not observed any dark exciton-related PL peak (Supplementary Fig. S12c, right). This can be explained with MoS₂ being in the top layer where the K-Q peak for Mo-based material then emerges at higher energy than MoS₂ A-exciton (as we observed for the MoSe₂ case: Fig. 1f) and this K-Q energy overlaps with inlayer WS₂ exciton. Another reason could be the energy of the Q-band, which is located far away from the K-band,¹⁴ subsequently incapable of band renormalization for K-Q excitons. The B-exciton of MoS₂ is also indistinguishable for the same overlapping energy with inlayer WS₂ exciton. We note that a similar PL peak to our S_Q peak has been observed previously in WS₂/MoS₂ heterobilayer.¹⁵

Fig. S12| Stacking sequence effect with other TMD combinations. **a**, PL spectra of WSe_2/MoS_2 and inverted MoS_2/WSe_2 heterostructure. Se_Q still emerges on the WSe_2 layer of WSe_2/MoS_2 heterostructure, whereas the S_Q peak in MoS_2 is rather ambiguous. **b**, PL spectrum of $MoSe_2/WSe_2$ and $WSe_2/MoSe_2$ heterostructure. The dark exciton peak was not visible due to overlaps with the bandgap. **c**, PL spectra of WS_2/MoS_2 and inverted MoS_2/WS_2 heterostructure. Se_Q still emerges on the WS_2 layer of WS_2/MoS_2 heterostructure, whereas the S_Q peak in MoS_2 is indistinguishable.

Comment 5: The authors were asked to give more analysis and explanation about the PL emission strength difference between A- and B-exciton of the MoS₂ from top-WSe₂/bottom-MoS₂ and top-MoS₂/bottom-WSe₂ heterostructures in Figure S10 a.

Response: We appreciate the reviewer for this interesting query. The excitonic properties of MoS₂ can be modified by complex many-body problems, for example, *charge transfer from proximate materials, substrate, strain, and more importantly the cleanness of the interface. Furthermore, the quantum yield of each material at the heterostructures matters as well.* In our case, the intensities of A and B – excitons of MoS₂ can be affected by the charge transfer from the proximate WSe₂ layer, hole traps from SiO₂/Si substrate, and possibly strained top layer. Consequently, charge screening is the key to determining the intensity of MoS₂. A-exciton intensity in MoS₂ can decrease due to increased charge screening of electrons, while the B-exciton intensity can increase due to charge-accumulated band filling.

Comment 6: Figure 4b gives the PL peak position mapping of top-WS₂/bottom-WSe₂ heterostructure, please add the corresponding PL intensity mapping.

Response: We thank the reviewer’s suggestion regarding the additional PL intensity mapping in the revised manuscript. Indeed, this intensity mapping gives a more detailed view of new PL peak distribution in real space. As shown in Fig. R1-13a, the integrated PL intensity of WS₂/WSe₂ heterostructure is plotted. Since the PL intensity of WS₂ is quite high compared to WSe₂, thus the WS₂ PL intensity is dominant compared to the heterostructure as well as WSe₂ region. To alleviate this difficulty, we normalized the S_Q intensity with respect to WS₂ PL intensity and plotted it in Fig. R1-13b. This gives the distribution of the S_Q peak over the heterostructure region, where the S_Q population is well represented compared to the intrinsic inlayer WS₂ intensity. The S_Q intensity is dominant at the heterostructure region as shown in Fig. R1-13b (yellow to the red region: scale >1). Furthermore, the intensity ratio changes from position to position, as described in Fig. 4c due to the variation of coupling strength between the layers.

Figure R1-13 (Fig. 4b and Supplementary Fig. S 24) | Intensity mapping. **a**, Integrated PL intensity of WS₂/WSe₂ heterostructure, where WS₂ PL intensity is dominant compared to the heterostructure as well as the WSe₂ region. **b**, Normalized S_Q intensity with respect to WS₂ PL intensity.

Considering this comment, we have included a new intensity mapping in Fig. 4 and modified our manuscript accordingly.

Modifications to the manuscript (page 7, line 176):

Inhomogeneous distribution of dark excitons. We investigate homogeneity of the K-Q excitons over the sample via PL energy and intensity mapping, as shown in Fig. 4 a,b (schematic and optical image in Fig. 4a top and inset). The light-bluish milky region with an energy range of 1.88 – 1.90 eV is referred to as K-Q excitons (Fig. 4a), which is inhomogeneously distributed over the entire heterostructure region. For clarity, we plotted the normalized S_Q peak distribution over the heterostructure region (details in Supplementary Note 18), where the S_Q population is dominant at the heterostructure region compared to the intrinsic inlayer WS_2 exciton intensity (Fig. 4b).

Fig. 4 | Theoretical modeling and application of Q-exciton as an optical power switch. a,b, PL energy and intensity mapping of WS₂/WSe₂ heterostructure at the box region shown in inset optical image of a. Inhomogeneous S_Q distribution over the heterostructure region. The S_Q intensity is dominant at the heterostructure region as indicated I_Q > I_K (arrow) in b. c, Extracted PL spectra, corresponding to A, B, and C positions: positive (I_Q - I_K) in A, which primarily covers the heterostructure region, I_Q - I_K ≈ 0 in B, and negative (I_Q - I_K) in C. d, Schematic demonstration of the Q-band renormalization depending on the strong, medium, and weak coupling at the interface. e,f, Experimental (top) and theoretical (down) photoluminescence for oppositely-stacked WS₂-WS₂: A compressive strain of respectively 0.3 and 0.15% in the top

layer leads to peaks corresponding to S_Q and Se_Q in the experimental data. **g**, Power-dependent PL spectra at high I_Q intensities with strong coupling region. **h**, In strong coupling regime, σ changes the polarity from positive to negative with increasing the laser power and acts as an optical power switch (red). This crossing point of K and Q- exciton intensity with laser power is clearly visible in the inset contour plot. In medium coupling regime, σ does not alter the polarity, although the power law is similar to that of strong coupling regime (green).

Modifications to the Supplementary (page 23, line 371):

Note 18. S_Q intensity mapping

PL intensity mapping may give a more detailed view of S_Q peak distribution in real space. As shown in Supplementary Fig. 24a, the integrated PL intensity of WS_2/WSe_2 heterostructure is plotted. Since the PL intensity of WS_2 is quite high compared to WSe_2 , the WS_2 PL intensity is dominant compared to the heterostructure as well as the WSe_2 region. To alleviate this difficulty, we normalized the S_Q intensity with respect to WS_2 PL intensity and plotted it in Supplementary Fig. 24b. This gives the distribution of the S_Q peak over the heterostructure region, where the S_Q population is well represented compared to the intrinsic inlayer WS_2 intensity. The S_Q intensity is dominant at the heterostructure region as shown in Supplementary Fig. 24b (yellow to the red region: scale >1). Furthermore, the intensity ratio changes from position to position, as described in Fig. 4c due to the variation of coupling strength between the layers.

Fig. S24 | Intensity mapping. a, Integrated PL intensity of WS_2/WSe_2 heterostructure, where WS_2 PL intensity is dominant compared to the heterostructure as well as the WSe_2 region. **b**, Normalized S_Q intensity with respect to WS_2 PL intensity.

Comment 7: As shown in Figure 1, the PL emission strength of WS₂ collected from WS₂/TMDs and TMDs/WS₂ heterostructures on SiO₂/Si substrate is much stronger than that of other TMDs. However, as shown in Extended Data Fig. 3 and Extended Data Fig. 5, these of WS₂/TMDs and TMDs/WS₂ heterostructures on quartz and HfO₂ substrates show opposite results. Please give explanation.

Response: This is a difficult question again due to many-body phenomena, as explained in Comment 5. In Fig. 1a,b, the substrate effect is dominant due to hole traps in SiO₂/Si substrate. In contrast, charge screening is reduced on quartz or HfO₂ substrate, and charge screening become dominant in the n-type WS₂ layer to reduce the PL intensity of WS₂. Nevertheless, this does not happen hBN substrate. A clear understanding of such charge screening effect is beyond our scope of the current work, which requires further investigation.

Comment 8: The authors are advised to add the absorption and time-resolved PL characteristics. In addition, first-principles calculations of band structures of heterostructures could be useful for explaining the origin of the new PL emission.

Response: We would like to thank the reviewer for providing additional insightful inputs, which we have taken into account and incorporated into our work. Moving forward, we discuss additional characteristics and theoretical modeling in the following section.

Part -1: Absorption measurement:

Measuring the indirect bandgap through the absorption spectrum in TMD heterostructures poses a challenging task due to the involvement of both direct and indirect bandgaps, along with the band renormalization effect including Q-exciton. This is well contrasted with PL, where indirect exciton also exhibits prominent features due to carrier transfer between different valleys (such as K to Q valley). Nevertheless, several studies have reported a slight variation in the absorption spectrum of TMDs heterostructure due to strong interlayer coupling [ACS Photonics 9, 1709-1716 (2022); ACS Nano 10, 6612-6622 (2016); Journal of the Optical Society of America B 35, 1179-1185 (2018)]. For example, the absorption peaks of MoS₂/WS₂ heterostructure (Fig. R1-14) are similar to those of ML MoS₂ and WS₂ but with a slight blueshift [ACS Photonics 9, 1709-1716 (2022)]. However, no indirect peak (Q-exciton) was observed unlike that in PL in our case.

Figure R1-14 | Previous study [Reference: ACS Photonics 9, 1709-1716 (2022)]. (a) Absorption spectra of 1L WS₂, 1L MoS₂, and WS₂/MoS₂ heterostructure.

Incorporating the reviewer's advice, we first measured absorption on the SiO₂ substrate (Fig. R1-15a) to make a direct comparison with our PL data. On the SiO₂ substrate, we observed additional defect-induced absorption peaks (grey peak from deconvolution in Fig. R1-15b) due to the coupling between hole trap charges in SiO₂ and excitons in TMDs, as reported in our previous study [ACS Nano 15, 2849-2857 (2021)]. We also found that each peak is redshifted in WS₂/MoS₂ heterostructures compared to monolayer peaks (Fig. R1-15a), analogous to the redshift observed in bilayer compared to monolayer, as reported previously [ACS Nano 7, 791-797 (2013); Nanomaterials 8, 725 (2018); Nanotechnology 27, 115705 (2016)], and this behavior has been attributed to the transition from direct to indirect bandgap.

For more detailed analysis, we deconvoluted each of the monolayers (Fig. R1-15b) as well as that of the heterostructure (Fig. R1-15c). In monolayer WS₂, we observed the WS₂ A-neutral exciton at ~1.99 eV, WS₂ trion at ~1.94 eV, and several defect-introduced excitons. Similarly, in monolayer MoS₂, we observed the MoS₂ A-exciton at ~1.88 eV and B-exciton at ~2.0 eV, as well as SiO₂ trap-related defect-induced excitons. In WS₂/MoS₂ heterostructure, we observed that each peak at the heterostructure red-shifted, and the trion of WS₂ increased significantly compared to neutral exciton due to band renormalization, as identified in deconvoluted spectrum in Fig. R1-15c. We note that no additional peak related to K-Q exciton was observed in this analysis and such an absence of K-Q peak from absorption originates from indirect nature. Obviously, we observe K-Q peak from PL, as shown in Fig. R1-15d. Such observed indirect K-Q exciton in PL is understood by the valley-to-valley carrier transfer in the conduction band from K to Q, and after that, recombined from Q in the conduction band to K in the valance band. A similar K-Q peak from absorption disappeared from the quartz substrate (Fig. e), although K-Q exciton in PL was still observed (Fig. f). Therefore, we conclude that K-Q peaks cannot be observed from absorption, congruent with previous reports as well [ACS Nano 7, 791-797 (2013)].

Figure R1-15 (Supplementary Fig. S19) | Absorption measurement. **a**, Absorption spectra of 1L WS₂, 1L MoS₂, and WS₂/MoS₂ heterostructure on SiO₂ substrate. **b,c**, Deconvolution of each monolayer as well as heterostructure absorption spectra. **d**, PL spectra of WS₂/MoS₂ heterobilayer on SiO₂. **e, f**, Absorption and PL spectra of WS₂/WSe₂ heterostructure on quartz substrate.

For this concern, we included the absorption part in the supplementary and added one more sentence in the revised manuscript.

Modifications to the Manuscript (page 5, line 124):

The emergence of these unusual peaks in PL can be explained in terms of the Q-band downshift in the top layer via band renormalization in the heterostructure region, which has been demonstrated previously for WS₂-WSe₂ hetero-bilayer⁸ and bilayer of MoSe₂.²⁷ In contrast, such peaks (S_Q/Se_Q) was absent in absorption measurements due to the indirect nature of S_Q or Se_Q peak (see details in Supplementary Note 13).

Modifications to the Supplementary (page 19, line 314):

Note 13. Absorption measurements

Measuring the indirect bandgap through the absorption spectrum in TMD heterostructures poses a challenging task due to the involvement of both direct and indirect bandgaps, along with the band renormalization effect. This is well contrasted with PL, where indirect exciton also exhibits prominent features due to carrier transfer between different valleys (such as K to Q valley). Nevertheless, several studies have reported a slight variation in the absorption spectrum of TMDs heterostructure due to strong interlayer coupling.^{18–20} However, no indirect peak (Q-exciton) was observed unlike that in PL.²¹

We first measured absorption on the SiO₂ substrate (Supplementary Fig. S19a) to make a direct comparison with our PL data. We found that each peak is redshifted in WS₂/MoS₂ heterostructures compared to monolayer peaks, analogous to the redshift observed in bilayer compared to monolayer, as reported previously,^{21–23} and this behavior has been attributed to the transition from a direct to an indirect bandgap. We note that no additional peak related to K-Q exciton was observed in this case and such an absence of K-Q peak from absorption may originate from indirect nature. Similar K-Q peak from absorption appeared from quartz substrate (Supplementary Fig. S19b) as well, although K-Q exciton in PL was observed (Extended data Fig. 5). Therefore, we conclude that K-Q peaks cannot be observed from absorption, congruent with previous reports as well.²¹

Fig. S19 | Absorption measurements. **a**, Absorption spectra of 1L WS₂, 1L MoS₂, and WS₂/MoS₂ heterostructure on SiO₂ substrate. **b**, Absorption spectra of 1L WS₂, 1L WSe₂, and WS₂/WSe₂ heterostructure on quartz substrate.

Part -II: Time-Resolved Photoluminescence (TRPL) measurement:

The reviewer advised to add the time-resolved photoluminescence (TRPL) measurements to our samples. Previously, the lifetimes of *interlayer* K-Q and K-K valley in heterostructure have been measured by TRPL spectroscopy; the decay time (a few hundred ps scales) of *interlayer* K-Q valley is longer than that of K-K valley (*Nature Materials*. 2020, 19, 617–623). In our case, we are concerned about the *intralayer* K-Q Valley. We expect to have a shorter lifetime. This fast TRPL system is not easily accessible. This requires further investigation. In our case, we observed the power dependence of *intralayer* K-K and K-Q peaks, as shown in Fig. 3. Since K and Q bands are at similar energy, the power exponents are not much deviated from each other. It strongly suggests that K-Q excitons are inherent from *intralayer* excitons similar to K-K exciton as our conclusion, which is distinct from *interlayer* exciton of previous reports.

Part-III: Theoretical calculation:

In our study we begin with first-principle calculation on the heterostructure, however, it always has identical results regardless of stacking sequential order. This arises from the absence of

symmetry breaking within the system. Therefore, to microscopically determine the excitonic energies we start from the unstrained single-particle dispersion [2D Mater. 2, 022001 (2015)] and use the strain-dependent variation of single-particle parameters provided in Ref. [2D Mater. 6, 015015 (2019)]. These values are then used to numerically solve the Wannier equation introducing a generalized Keldysh potential for the Coulomb interaction. As a result, we find that the K-Q exciton is located below the K-K exciton for W-based materials, while the opposite is the case for MoSe₂, (see below). In view of the different behaviors of K-K and K-Q excitons in the presence of strain, applying compressive strain results in an increased energy separation between K-Q and K-K states in W-based materials, the blue line. This allows for the appearance of the peaks Se/S_Q driven by the larger occupation of K-Q excitons.

Figure R1-16 (Supplementary Fig. S27) |Dispersion of dark excitons. Energy of K-Q excitons as a function of the center-of-mass momentum for unstrained (blue) and compressively strained (red) **a**, WS₂ and **b**, MoSe₂.

Modifications to the Manuscript (page 17, line 460):

In this way we microscopically predict in **W-based materials** a strain-induced increase of the energy separation between bright K-K and momentum-dark K-Q excitons, with a consequent increase in the corresponding occupations. This in turn results in an increased indirect emission from monolayer K-Q states in comparison to the direct one, as shown in Figs. 4e,f. In particular, the strain in the top layer induces new peaks in good agreement with the experimental observation. **The phonon-assisted emission has a lower efficiency than direct recombination, which is, however, compensated by a higher occupation of the K-Q excitons compared to the bright excitons. The higher relative occupation of K-Q excitons is induced by the energy alignment in tungsten-based materials and is further triggered by compressive strain. Furthermore, at decreased temperatures (77 K), we find an increased relative occupation of K-Q excitons and hence more intense phonon-sidebands.³¹ In contrast, in Mo-based materials, K-Q excitons are higher in energy than the bright-excitons even in the presence of compressive strain, Supplementary Fig. 27. This leads to a negligible phonon-assisted PL. Nevertheless, the strain-induced reduction of the dark-bright exciton separation (Supplementary Fig. 28) could potentially lead to a resonance similar to Se_Q in MoSe₂/WS₂, if additional activation mechanisms are present, e.g., via the interplay of strain and defects.³⁷ The microscopic evaluation of these mechanisms goes beyond the scope of this work.**

Reviewer #2:

General comment:

In the manuscript entitled “Sequential order dependent dark-exciton modulation in bi-layered TMD heterostructure”, Sebait et al studied how stacking sequence affect momentum-dark exciton emission in TMDs heterobilayers. Since most of previous studies focus on understanding the optical properties of interlayer exciton states, it is quite interesting to see a detailed study on exploring intralayer exciton states. Although the authors present some experimental evidence combined with theoretical modeling to support their conclusions, I still have a few concerns on how they interpret their results. The authors need to fully address my following comments before I can recommend to publish on Nature Communications.

Response: We are pleased to hear that reviewer feels interested in this work, as it is rather different from the previously studied interlayer exciton. We thank the reviewer to bring numerous valuable comments to significantly improve our revised manuscript accordingly. Please see the point-by-point response below.

Comment 1: In the Fig. 1a, the authors observe a new emission state at 1.89 eV in the WS₂(top)/WSe₂(bottom) heterobilayer which they attribute to the indirect exciton (K-Q) emission. However, a few groups previously have fabricated similar samples and they don't see such features (e.g., ACS Nano 2016, 10, 6612–6622, Small 2019, 15, 1902424). How the authors explain the discrepancy?

Response: We thank the reviewer for raising the concerns regarding the previous studies where no such K-Q dark exciton is observed. We are also thankful to the reviewer for giving us the opportunity for clarifying the concerns. To the best of our knowledge, the fabrication process of the heterobilayer is one of the key factors for observing such K-Q dark excitons, because the strong interlayer coupling is deemed necessary for observing such phenomena. Furthermore, K-Q excitons are also dependent on the materials of the heterostructure. We will discuss here the previous reports that do show K-Q excitons unintentionally for given strong interlayer coupling, whereas such K-Q excitons are not observed in the case of weak interlayer coupling. In our case, we paid attention to having strong interlayer coupling during heterostructure fabrication with different materials, during transfer without involving residues. Therefore, firstly we discussed the fabrication process and stacking sequence details and then compared with the reported results one by one.

(A) Strongly coupled clean interface:

We used aligned dry-transfer technique for stacking the selected flake on another designated flake. The top layer was first exfoliated on PMMA/PVA-coated substrate. A customized holder with a 2-3 mm hole was used to scoop the TMD/PMMA film after dissolving the PVA in warm water. The bottom layer was directly exfoliated onto the substrate and aligned with the top TMD/PMMA and stacked together. The transfer temperature of 110 °C was maintained during staking to avoid interfacial residues. The stepwise stacking procedure was schematically shown in Supplementary Fig. S1 and discussed in detail in the method section (page 15). Figure R2-1a shows the K-Q exciton from the clean interface sample. Meanwhile, in another

heterostructure, where the bottom layer was transferred with polymer (such as PMMA) to the target substrate, and then cleaned the polymer by using the conventional acetone-IPA-ethanol cleaning process, followed by the top layer was transferred on top of the bottom layer. Even though polymers were cleaned, some unwanted contaminants still remained, and a clean surface can be hardly achieved. As a consequence, we did not find any clear signature of Q-band-related dark exciton peak (S_Q) in this case (Fig. R2-1b). Meanwhile, the cleanliness-dependent strong coupling has been confirmed by the presence of A^2_{1g} Raman mode of WSe_2 (Supplementary Fig. S5) exclusively at the clean heterobilayer region. A homogeneous topographic image by AFM from the clean heterobilayer region was also confirmed in Supplementary Fig. S6. The detail has been described in supplementary information note 3 (SI page 5).

Fig. R2-1 (Supplementary Fig. S4) | Interface cleanliness. **a**, Schematic representation of WS_2/WS_2 heterostructure; A clear S_Q peak was observed from the clean interface by our transfer method. **b**, Schematic representation of contaminated heterostructure; At the contaminated interface, the S_Q peak was invisible.

Considering the reviewer's comments, we added the above schematic in the Supplementary Fig. S4.

(B) Comparison with previous reports:

(i) ACS Nano 2016, 10, 6612–6622:

Stacking sequential order: WSe_2 (top layer)/ WS_2 (bottom layer) (main manuscript) and WS_2 (top layer)/ WSe_2 (bottom layer) (Supplementary)

Fabrication method: Heterostructures were prepared by the PMMA stamping method. First, PMMA was coated on the CVD-grown WS_2 flake grown on a sapphire substrate. The WS_2 flake was detached from sapphire by KOH solution and thereafter transferred to ML WSe_2 flake at 100 °C. Heterostructures were then annealed at 350 °C to remove the polymer residues.

Comparison of the results: Authors never observed any clear interlayer exciton peak due to interfacial trap/residue between the layers (as shown in the AFM image in supplementary Fig. S4 and here Fig. R2-2a). The interlayer coupling has been improved via thermal annealing, which makes efficient charge separation and enhancement of light absorption at heterostructure.

In $\text{WSe}_2(\text{top})/\text{WS}_2(\text{bottom})$ heterostructure, a small additional PL peak (~ 830 nm) was observed at the heterostructure (marked circle in Fig. R2-2b). This peak vanishes in the reverse stacking case (Fig. R2-2c). However, the new peaks were not that prominent due to remaining interfacial residues but still had a significant signature that matches well with our observation.

Figure R2-2 | Comparison with previous work. a, AFM mapping demonstrates the removal of interfacial contamination before and after annealing. **b,c,** PL spectra of WSe_2/WS_2 and inverse (WS_2/WSe_2) heterostructure.

(ii) Small 2019, 15, 1902424:

Stacking sequential order: WSe_2 (top layer)/ WS_2 (bottom layer)

Fabrication method: The WS_2 was directly exfoliated on a substrate (SiO_2) and WSe_2 was exfoliated on polymethyl-methacrylate (PMMA) and followed by transferring the WSe_2 layer on top of the WS_2 layer via dry transfer technique. Subsequently, the as-prepared WSe_2/WS_2 hetero-bilayers (HBs) were annealed in an Ar atmosphere (3 Torr, 200 Scm) at 300°C for 12 hours to enhance the vdW interaction between the layers. The procedure is considered a clean interface.

Comparison of the results: Authors observed interlayer exciton around 1.5 eV at low temperatures ($<100\text{K}$) (Fig. R2-3a) and close to 1.55 eV at room temperature. This interlayer exciton energy difference from inlayer WSe_2 exciton was around 60-80 meV (shown in Fig. R2-3a inset), different from other previous reports [Nature 2019, 567, 76; Optical Express 2020, 28, 13260; Nature Materials 2020, 19, 617]. Such an energy difference of interlayer exciton matches similarly with our K-Q exciton (Se_Q peak in Fig. R2-3b). We like to emphasize that interlayer exciton strongly depends on angle alignment

between the layers as described previously [Nature 2019, 567, 76; Nature 2022, 610, 478; Nature Materials 2020, 19, 617; Nature Communications 2020, 11, 5888; ACS Nano 2017, 11, 4041]. However, in their work authors did not mention the angle alignment between the layers. In other words, the observation of interlayer exciton in their case is rather ambiguous.

Figure R2-3 | Comparison with previous work. **a**, Top: Schematic and optical image of WSe_2/WS_2 heterobilayer. Bottom: PL spectra at WSe_2/WS_2 heterobilayer region, an additional peak emerges at the heterostructure region called an interlayer exciton (X_{IL}) [Reference: Small 2019, 15, 1902424]. **b**, Our results for a similar stacking configuration to **a**, an additional peak at heterostructure (cyan color), which is called Se_Q peak emerging due to the K-Q intralayer excitons.

(C) Comparison with previous reports (additional):

While the reviewer suggested to the correlation of the previous references from K-Q excitons, we like to list a few more reports (some of them suggested by Reviewer 1) to provide PL evidence of our K-Q exciton at heterostructure depending on the stacking sequential order.

(iii) Nature 2022, 610, 478:

Stacking sequential order: WS_2 (top layer)/ WSe_2 (bottom layer) (main manuscript) and WSe_2 (top layer)/ WS_2 (bottom layer) (Supplementary).

Fabrication method: Bottom WSe_2 layer was dry-transferred from the gel film to the substrate first. The second layer of WS_2 was then transferred with a thermal heating process at 363 K. Heating the layers improves the interlayer coupling between them. Therefore this procedure is considered a clean interface.

Comparison of the results: Authors observed several emissions due to moiré-trapped interlayer excitons in suspended WSe_2/WS_2 heterobilayer. Reduced screening in free-standing heterobilayer, led to more robust dipole-dipole interactions, that confirm the visibility of interlayer excitons. They observed the nature of the dipole interactions among free interlayer

excitons; from repulsive to attractive, caused by quantum-exchange-correlation effects, which lead to the appearance of interlayer biexciton phases.

Further, the authors demonstrated the PL spectra on both WSe_2/WS_2 and WS_2/WSe_2 heterobilayer in Fig. R2-4 (see also their Extended Data Fig. 3b and c). As shown in Fig. R2-4a, they observed some additional PL peak (red box area) below the inlayer WS_2 peak (similar to our S_Q peak), whereas such peak was not observed in the reverse stacking sequence of WSe_2/WS_2 heterostructure (Fig. R2-4b). On the other hand, in Fig. R2-4b, there was another additional peak (similar to our Se_Q peak) below WSe_2 inlayer exciton (marked in red circle), again absent in the alternative stacking case as shown in Fig. R2-4a. Nevertheless, the authors did not pay attention to these results and never explained origin of such peaks, which could be related to K-Q excitons in our study.

Figure R2-4 | Comparison with previous work. (a,b) PL spectra of WS_2/WSe_2 and WSe_2/WS_2 heterobilayer, an additional peak emerges below the top layer exciton similar to our S_Q and Se_Q peak respectively [Reference: *Nature* 2022, 610, 478]. The red box and circle have been drawn to represent the peak position, which is present or absent depending on stacking sequential order.

(iv) Nano Lett. 2014, 14, 3185-3190:

Stacking sequential order: WS_2 (Top layer)/ MoS_2 (bottom layer) (main manuscript) and MoS_2 (top layer)/ WS_2 (bottom layer) (Supplementary).

Fabrication method PDMS was spin-coated on CVD-grown monolayer WS_2 and PDMS/ WS_2 was then separated from SiO_2/Si substrate by KOH solution and further transferred to monolayer MoS_2 via PDMS stamping techniques. Further, the cleanliness of the interface was improved by annealing the heterostructures.

Comparison of the results: Authors demonstrated the tuning of interlayer coupling via thermal annealing. In the strongly coupled region, authors found two additional peaks along with inlayer excitonic peaks, named – P_{hetero} and P_{indirect} at energy 1.94 and 1.75 eV respectively

(Fig. R2-5a). The P_{indirect} peak position changes rapidly with annealing time and is attributed to phonon-assisted indirect band gap transition. The P_{hetero} peak position is ~ 60 meV below the inlayer WS_2 peak, and this amount of energy shift is lower than the accuracy of DFT calculations. Further, authors discussed the P_{hetero} emission in terms of excitonic effects and ultrafast charge relaxation across the heterostructure. However, it does not explain the distinct behavior of MoS_2 , leaving the origin of the P_{hetero} emission elusive.

In Supplementary data, on the other hand, authors also fabricated the reverse stacking ($\text{MoS}_2(\text{top})/\text{WS}_2(\text{bottom})$) and in this case, surprisingly the P_{hetero} peak vanishes (Fig. R2-5c). Authors did not pay attention to this as it is very unconventional. This report strongly supports our work. For clear demonstration, we also fabricated a similarly configured heterostructure and measured the PL. WS_2/MoS_2 heterostructure reveals the S_Q peak (which authors named P_{hetero} in their work), while MoS_2/WS_2 heterostructure vanishes the S_Q peak (Fig. R2-5b,d). This clearly demonstrates that the origin of the K-Q exciton peak is determined by the stacking sequential order.

Figure R2-5 | Comparison with previous work. a,b, PL spectra of WS_2/MoS_2 heterostructure of previously reported [Reference: *Nano Lett.* 2014, 14, 3185-3190] and our result. c,d, a similar result for inverse stacking MoS_2/WS_2 .

(v) *Nat. Phys.* 2018, 14, 801:

Stacking sequential order: WSe₂ (top layer)/MoS₂ (bottom layer)

Fabrication method: Both MoS₂ and WSe₂ exfoliated onto polydimethylsiloxane (PDMS) substrates. First MoS₂ flake was transferred onto the target substrate. Subsequently, the WSe₂ flake was transferred on top of MoS₂. Afterward, annealing the heterostructure at 150° C for 5 hrs in an H₂/Ar gaseous environment. This annealing process promotes strong interfacial coupling, resulting in the clean interface.

Comparison of the results: This article also reported an additional exciton peak only at the heterostructure region at the energy of 1.6 eV, 60-70 meV lower than WSe₂ neutral exciton (Fig. R2-6a). Authors claim that this additional peak appeared due to interlayer exciton between WSe₂-MoS₂ heterobilayer. In this case, interlayer exciton always appeared independent of the stacking angle, in contrast to other previous reports [Nature 2019, 567, 76; Nature 2022, 610, 478; Nature Materials 2020, 19, 617; Nature Communications 2020, 11, 5888; ACS Nano 2017, 11, 4041]. Further, the intensity at room temperature was comparable to or higher than that of neutral exciton of each layer, again distinct from previous reports [Phys. Status Solidi B 2019, 256, 1900308; Nature 2019, 567, 76; Nature 2022, 610, 478; Small 2019, 15, 1902424; Nature Materials 2020, 19, 617; Optics Express 2020, 28, 13260]. This cannot be explained by non-equilibrium effects alone. The authors justified the reason by considering Γ -K indirect transition instead of K-K interlayer transition.

This additional peak energy below the inlayer WSe₂ peak is also similar to that of the K-Q exciton (Se_Q peak) (Fig. R2-6b) in our case. However, experimentally we observed that additional peak (Se_Q peak) vanishes for altering the stacking sequential order (Supplementary Fig. S12a), even though other parameters remain identical. Furthermore, the authors mentioned that the additional PL peak explained by interlayer exciton is independent of the stacking angle, which is again similar to our finding. Authors claim that this additional peak is due to Γ -K interlayer exciton, which is different from our claim.

Figure R2-6 | Comparison with previous work. a, PL spectra at WSe₂/MoS₂ heterobilayer region, an additional peak emerges below the inlayer WSe₂ exciton [Reference: *Nat. Phys.* 2018, 14, 801]. **b,** Our results, similar to **a**, also had the same additional peak at heterostructure, which we called Se_Q peak emerging due to the K-Q intralayer excitons.

(vi) **Phys. Status Solidi B** 2019, 256, 1900308:

Stacking sequential order: MoS₂-WSe₂ and MoSe₂-WSe₂. Any specific stacking sequential order has not been mentioned.

Fabrication method: TMD flakes were exfoliated onto PDMS substrate from bulk crystal, subsequently transferring the constituent flakes of a heterostructure onto the target substrate. MoS₂-WSe₂ heterostructure was randomly oriented to each other, whereas crystallographic angles were aligned in MoSe₂-WSe₂ heterostructure. This is considered a strong coupling.

Comparison of the results: Authors presented the experimental and theoretical studies of interlayer exciton in two different heterobilayer combinations such as twisted MoS₂-WSe₂ and aligned MoSe₂-WSe₂. In both cases, interlayer excitons have been observed. However, these interlayer excitons show different properties. In the MoS₂-WSe₂ heterostructure, interlayer exciton was observed near 1.6 eV, 60-70 meV below the WSe₂ inlayer exciton, and this interlayer exciton is independent of the stacking angle between the layers (Fig R2-7a). Further, authors claim from theoretical calculations that this interlayer exciton appears due to indirect Γ -K transition instead of K-K interlayer transition. On the other hand, in the MoSe₂-WSe₂ heterostructure, they got interlayer exciton of more than 200 meV separated from residual monolayer emission (Fig R2-7b). In this case, authors claim that it is a K-K transition and strongly depends on the twisted angle between the layers. Although authors did not clearly indicate the stacking sequential order, which is an important parameter to compare with our result. We presume that the WSe₂ layer is located on the top layer of the MoS₂ layer as per their previous report [Proc. Natl. Acad. Sci. 2014, 111, 6198].

Figure R2-7 | Comparison with previous work. a,b, PL spectra of WSe₂/MoS₂ and WSe₂/MoSe₂ heterobilayer, an additional peak emerges at the heterobilayer region called interlayer excitons (ILE) [Reference: *Phys. Status Solidi B* 2019, 256, 1900308]. c,d, Our results with similar stacking configuration to a,b.

In our approach, in the top WSe₂/bottom MoS₂ heterostructure (Fig R2-7c), we observe a new additional PL peak (K-Q peak) below the inlayer WSe₂ peak. We interpret our peak as K-Q intralayer exciton. If we reverse the stacking sequential order to the top MoS₂/ bottom WSe₂ heterostructure, such K-Q exciton disappears, which is well distinguished from interlayer exciton in their work. In the case of MoSe₂-WSe₂ heterostructure, we have never observed any Se_Q peak due to overlapping the energies between two materials as we have mentioned in our manuscript (Supplementary Fig. S12b). However, it would be noted that in our case we have also observed a tiny signal of interlayer exciton around the energy 1.35 eV, similar to this mentioned report at room temperature (inset of Fig R2-7d).

(vii) Optics Express 2020, 28, 13260

Stacking sequential order: WSe₂ (top layer)/WS₂ (bottom layer).

Fabrication method: Each of the monolayer flakes has been exfoliated from the bulk crystal onto the silicon substrate. After that, the heterostructure was constructed by the fixed point transfer technique. However, the fabrication details have not been provided in this work.

Comparison of the results: Authors demonstrated the existence of double indirect interlayer exciton in WSe₂/WS₂ heterobilayer at the cryogenic temperature at energy 1.4585 eV and 1.4885 eV. At room temperature, authors observed a single interlayer exciton below the inlayer WSe₂ peak (Fig. R2-8a). Their interlayer exciton peaks are similar to that of our K-Q excitons (Se_Q peak in Fig. R2-8b), although our interpretation is very different from theirs. We emphasize that we confirmed Se_Q peaks from stacking sequential order.

Figure R2-8 | Comparison with previous work. a,b, PL spectra of WSe₂/WS₂ heterostructure of previously reported [Reference: *Optics Express* 2020, 28, 13260] and our result. The red dotted circle indicates the emergence of an additional peak at the heterostructure region.

In conclusion, the comparisons with previous reports are summarized below:

- (i) Optical anomalies of emergence or disappearance of exciton peaks have been observed recently from few reports that depend on stacking sequential order [Nature 2022, 610, 478; Nano Lett. 2014, 14, 3185-3190; ACS Nano 2016, 10, 6612–6622]. However, experimental verification of these anomalies is always ignored.
- (ii) Different combinations of TMDs generate distinct interlayer excitonic peaks, but previous reports have shown almost the same energy for different combinations, for example, WSe₂-WS₂ [Optics Express 2020, 28, 13260; Small 2019, 15, 1902424] and WSe₂-MoS₂ [Nature 2018, 560, 340; Phys. Status Solidi B 2019, 256, 1900308; Nat. Phys. 2018, 14, 801].
- (iii) Interlayer exciton should be independent of stacking sequential order [Nature 2022, 610, 478–484; Nat. Commun. 2015, 6, 4–9]. However, we have observed that by altering the stacking sequential order exclusively and keeping all the other parameters (experimental conditions) the same, our heterostructure peak emerges or vanishes depending on stacking sequential order. To date, there is no such report on this unusual phenomenon.

In our opinion, all these previous findings are actually supporting our results. For example - if WSe₂ is the top layer in a hetero-bilayer, we will then have a strong K-Q intralayer peak corresponding to WSe₂, independent of the bottom layer. All such inconsistencies arise from the absence of no such systematic study that extensively investigates a variety of different TMDs combinations.

We thank again to the reviewer for this query, which provided a more clear picture of our findings and we have added the above-mentioned references in the revised version of the manuscript. In addition, we are obliged to modify the introduction accordingly due to a few references mentioned by the reviewer (Ref 11 added in the revised manuscript).

Modifications to the manuscript (page 2, line 28):

In addition, they also show a variety of optically forbidden dark excitons, either due to spin-flip or momentum transfer.⁵⁻⁷ For example, both K-K bright and momentum-forbidden K-Q excitons emerge in monolayer WSe₂, as confirmed by time- and angle-resolved photoemission spectroscopy.⁶ These momentum-forbidden indirect excitons can also be realized in van der Waals (vdW) heterostructures, where bound electrons and holes are localized in two different layers, which is called an interlayer exciton.⁷⁻¹⁴ Emergence of these indirect interlayer excitons strongly depends on the stacking angle between the layers^{8,10,12} and is independent of stacking sequential order.^{10,15} Despite this, a few recent reports have accidentally observed optical anomalies in heterostructures that rely on the sequential order of the layers.^{15,16} However, the underlying mechanisms that give rise to these anomalies are not yet fully realized. This lack of understanding may be attributed in part to the hypothesis that altering the stacking sequence between layers would not significantly affect the intrinsic properties of heterostructures. Therefore, it is important to explore stacking sequential order dependence to further use of real device applications.

In this work, we tune dark excitons by changing the stacking sequence between TMD layers, whereas other experimental parameters remain constant. Such emergence or disappearance of dark excitons depending upon stacking sequence has not been reported to date.

Comment 2: In the Fig. 2c, the authors claim the momentum dark K-Q exciton states (SQ) lie ~ 100 meV lower than K-K excitons. According to the Boltzmann equation ($(N(\text{bright exciton}))/N(\text{dark exciton}))=e^{(-\Delta E/k_B T)}$, ΔE is the energy difference between K-K and K-Q excitons), most of photo-excited excitons will populate in the energetically favorable K-Q states even at room temperature. So why direct exciton (K-K) recombination still contributes significantly in the PL emission as shown in the Fig. 1a and Extended Fig. 2?

Response: The referee raises an important point, as the photoluminescence in TMD monolayers and related heterostructures is dictated by a non-trivial interplay of weakly-populated bright states and highly populated dark states. The absorption is dominated only by the former bright states, i.e., those which interact directly with the light. Dark exciton states affect the absorption only indirectly via an increased spectral linewidth and asymmetric peak shape [Phys. Rev. Lett. **119**, 187402 (2017)]. In contrast, dark excitons can appear in PL spectra via phonon-assisted processes. Here, dark excitons first scatter into virtual states in the light cone (driven by the emission of acoustic and optical phonons), before then they recombine and emit light leading to the formation of phonon sidebands. This two-particle process is much less efficient than the direct recombination of the bright states. However, the low-energy dark excitons are much more populated than the high-energy bright states, and this overpopulation compensates for the smaller efficiency of the phonon-assisted recombination. As a result of this interplay between the high dark-exciton population and the low efficiency of the phonon-assisted emission, phonon sidebands can have a comparable intensity as the bright exciton.

In the revised version of the manuscript, we added additional text addressing the height of the S/Se_Q peak in comparison to the bright exciton peak, in particular stressing the delicate competition between occupation and efficiency of direct and phonon-driven exciton recombination.

Modifications to the manuscript (page 17, line 460):

This in turn results in an increased indirect emission from monolayer K-Q states in comparison to the direct one, as shown in Figs. 4e,f. In particular, the strain in the top layer induces new peaks in good agreement with the experimental observation. The phonon-assisted emission has a lower efficiency than direct recombination, which is, however, compensated by a higher occupation of the K-Q excitons compared to the bright excitons. The higher relative occupation of K-Q excitons is induced by the energy alignment in tungsten-based materials and is further triggered by compressive strain. Furthermore, at decreased temperatures (77 K), we find an increased relative occupation of K-Q excitons and hence more intense phonon-sidebands.³¹ In contrast, in Mo-based materials, K-Q excitons are higher in energy than the bright-excitons even in the presence of compressive strain, Supplementary Fig. 27. This leads to a negligible phonon-assisted PL. Nevertheless, the strain-induced reduction of the dark-bright exciton separation (Supplementary Fig. 28) could potentially lead to a resonance similar to Se_Q in

MoSe₂/WS₂, if additional activation mechanisms are present, e.g., via the interplay of strain and defects.³⁷ The microscopic evaluation of these mechanisms goes beyond the scope of this work.

Comment 3: In the Extended Fig. 2, the authors measure a few samples with different twist angles. It is interesting to see the emission intensity and energy of SQ change with twist angle. A previous study has shown the twist angle could significantly affect intralayer exciton states in WS₂-WSe₂ heterobilayer, specifically, the energy difference between K-Q and K-K excitons (Nature Materials. 2020, 19, 617–623). The authors need to carry out more careful analysis on the twist-angle-dependent behavior as it provides another solid evidence for their interpretations.

Response: We appreciate the reviewer for these valuable comments. We agree with the reviewer's comments that in Extended data Fig. 2, emission energy and intensity of S_Q or S_{eq} changes from sample to sample, and one may conclude that it may vary with the twist angle. However, in a real scenario, it is a bit complicated, as shown in PL mapping across the heterostructure region (Fig. 4 a,b), S_Q peak energy and intensity vary from position to position. This phenomenon can be explained in terms of coupling strength or interlayer hybridization between the layers, which has been described in Fig. 4, section '*Inhomogeneous distribution of dark exciton*'.

The mentioned reference paper (Nature Materials. 2020, 19, 617–623) demonstrated the twist-angle-dependent interlayer exciton diffusion in WS₂-WSe₂ hetero-bilayers. Since the mentioned work focuses on *interlayer exciton* and thus it depends on the *stacking angle between the layers*, similar to other previous work [Nature 2019, 567, 76; Nature 2022, 610, 478; Nature Communications 2020, 11, 5888; ACS Nano 2017, 12, 4041]. However, the essence of our work is different, as it focuses on *intralayer exciton* and we have observed that *it does not depend on the stacking angle between the layers* (newly added Supplementary Fig. S9 and Table S1). Besides, in that paper authors claim that instead of conventional K-K interlayer exciton, K-Q was the ground state, and they also calculated the energy difference between K-Q (ΔE_{K-Q}) is 88-62 meV, which perfectly fits with our result, i.e., the peak energy difference between K-K and K-Q intralayer exciton is 90-60 meV and K-Q resides at lower energy than K-K exciton. Furthermore, their DFT calculation of Moiré potentials demonstrated that the potential varies from position to position, that effect the ΔE_{K-Q} , which again reflects in our result as well (Fig 4b PL mapping).

To address the reviewer's suggestion, we performed second harmonic generation (SHG) measurements on seven samples to determine the stacking angle between the layers (Table R2-1), where we observed the indirect K-Q peak. One of those has been illustrated in Fig. R2-9. Based on this observation, we may conclude that the appearance of our newly observed K-Q intralayer peak is independent of the stacking angle, making it more robust compared to moiré or interlayer excitons.

This demand that our observed intralayer K-Q exciton has great potential for future optoelectronic applications, as it is indirect and independent of the stacking angle and more

importantly, PL intensity similar to monolayer intensity at room temperature. We thank the reviewer for this suggestion, it impacts greatly improves our manuscript.

Figure R2-9 (Supplementary Fig. S9) | Twist-angle measurements using second-harmonic generation (SHG). a, Optical image of WS₂/WSe₂ heterobilayer. b,c, Polarized SHG intensity polar diagram as a function of incident field polarization angle for WS₂ and WSe₂, respectively. The black dots correspond to experimental data and the red line is the fitting, which reveals the stacking angle between this two-layer is -7.09°.

Table R2-1 (Supplementary Table S1) | Stacking angle measured via second harmonic generation (SHG).

Sample name	Stacking sequence (top/bottom) layer	Stacking angle (degree)
RD13	WS ₂ /WSe ₂	16.36
RD14	WSe ₂ /WS ₂	15.14
RD26	WS ₂ /WSe ₂	8.48
RD51	WS ₂ /WSe ₂	24.53
RD52	WS ₂ /WSe ₂	24.31
RD64	WS ₂ /WSe ₂	-7.09
RD73	WS ₂ /WSe ₂	35.34

Modifications to the manuscript (page 3, line 73):

We note that all heterostructures were randomly stacked and the stacking angle between the layers was further confirmed by second harmonic generation (SHG) measurements (Supplementary Fig. 9). Therefore, the observed features are not specific to a stacking angle.

Modifications to the Supplementary (page 9, line 139):

Note 5. Stacking angle measurements

We performed second harmonic generation (SHG) measurements on seven different samples to determine the stacking angle between the layers (Table S1), where we observed the indirect K-Q peaks, as illustrated in Supplementary Fig. 9. Based on the observations, we conclude that

our newly observed K-Q intralayer peak is independent of the stacking angle, making it more robust compared to moiré or interlayer excitons.

Fig. S9 | Twist-angle measurements using second-harmonic generation (SHG). **a**, Optical image of WS₂/WSe₂ heterobilayer **b,c**, Polarized SHG intensity polar diagram as a function of incident field polarization angle for WS₂ and WSe₂ respectively. The black dots correspond to experimental data and the red line is the fitting, which reveals the stacking angle between two layers is -7.09°.

Table S1 | Stacking angle measured via second harmonic generation (SHG).

Sample name	Stacking sequence (top/bottom) layer	Stacking angle (degree)
RD13	WS ₂ /WSe ₂	16.36
RD14	WSe ₂ /WS ₂	15.14
RD26	WS ₂ /WSe ₂	8.48
RD51	WS ₂ /WSe ₂	24.53
RD52	WS ₂ /WSe ₂	24.31
RD64	WS ₂ /WSe ₂	-7.09
RD73	WS ₂ /WSe ₂	35.34

Comment 4: Finally, the authors theoretically explain the possible origin of K-Q dark exciton emission is due to top layer is more strained than the bottom layer. However, in the stacked heterobilayers, the formation of nanobubbles is unavoidable which could induce localized strain regions with exciton localization in both WS₂ and WSe₂ layers (Nature Nanotechnology, 2020, 15, 854–860). Moreover, as shown in the PL mapping (Fig. 4b), it could explain the inhomogeneous distribution of this new emission state. How the authors rule out this possible cause?

Response: We thank the reviewer for the critical revision of our work, which has greatly contributed to its refinement. The reference paper provided the additional peak in the bubble area in monolayer WSe₂ on the hBN substrate. As can be seen below from the reference paper (Fig. R2-10), additional peak below inlayer WSe₂ exciton due to localized strain effect from nanobubbles area.

Figure R2-10 | Previous report about the localized exciton from nano-bubbles [Reference: Nature Nanotechnology, 2020, 15, 854–860]. **a**, Comparison of PL spectra collected from a flat and nano-bubble region. **b**, AFM topographic image of WSe₂/h-BN heterostructure. **c**, Special map of the localized exciton.

Here, we have systematically addressed one by one how we have ruled out that the origin of the K-Q exciton is not due to nanobubbles in our case –

(i) Comparison of PL spectra collected from the nano-bubble and flat surface: To confirm the effect of nanobubbles at the heterostructure in our case, we performed PL mapping on WS₂/MoSe₂ heterobilayer, which contains several bubbles (indicated by the arrow in Fig. R2-11 a,b) at the interface. We found that on top of the nanobubble (marked as B), the WS₂ inlayer peak underwent a red shift due to strain, while the MoSe₂ peak remained unaffected (blue spectra in Fig. R2-11 c). However, on the flat surface (marked as A), we observed a prominent and highly intense S_Q peak (orange spectra in Fig. R2-11 c). Moreover, the relative intensity of the bottom MoSe₂ layer increased at the flat surface, indicating strong coupling between the layers, which is deemed necessary for a high S_Q peak. This result reveals that the origin of the K-Q peak is not related to nano-bubbles at the interface.

Figure R2-11 (Supplementary Fig. S13) | Effect of nano-bubble. **a**, Optical image of WS₂/MoSe₂ heterobilayer. Nano-bubbles are indicated by arrows. **b**, Spatial PL map of WS₂/MoSe₂ heterobilayer, revealing that bright spots are localized in spatially discrete regions that correspond to the presence of nano-bubbles. **c**, Comparison of PL spectra collected from a flat (labeled as A: orange) and a nano-bubble region (labeled as B: blue). **d,e**, Deconvolution of the spectrum from the flat and nano-bubble region.

(ii) **The indirect peak at intact bilayer:** Intact homogeneous bilayers, which exhibit no bubbles or residue at the interface, show similar peaks (Fig.1 g,h) to those observed in our heterostructure (Fig. 1a-f).

(iii) **Post-annealing effect:** Nanobubbles can be removed by post-annealing as previously reported by several groups including the authors of the reference paper group (2D mater, 2018, 5, 031001; 2D mater, 2017, 4, 021019). In our case, (i) we fabricated our heterostructure at a high temperature (>110° C) and (ii) further post-annealing for 12 hours at 250 °C. As-fabricated samples may contain nanobubbles and polymer contaminations, whereas polymer contaminations and nanobubbles are largely removed after post-annealing. The new peak S_Q/Se_Q remains intact before and after annealing, as shown in supplementary Fig. S17. This again confirms that this additional peak is not related to nanobubble. Furthermore, a strong S_Q peak was observed on a clean interface compared to the contaminated interface as discussed in Supplementary Fig.S4.

We appreciate the insightful query from the reviewer. In response, we have incorporated a new figure in the supplementary along with a corresponding paragraph to elucidate the effect of nano-bubbles and our observed K-Q peak.

We have included the mentioned reference paper in our revised manuscript, reference no 23.

Modifications to the manuscript (page 4, line 105):

Furthermore, the exfoliated bilayer interface remains intact and hence the presence of such an unusual peak reassures the clean interface in our fabricated hetero-bilayers (Supplementary Fig.

4). We also observed strong and distinct S_Q peak on the spatially uniform and bubble-free flat heterostructure surface (Supplementary Fig. 13), which differs from previously reported localized exciton due to the presence of bubbles.²³

Modifications to the Supplementary (page 15, line 213):

Note 9. Effect of nano-bubbles

To distinguish between the localized excitons originating from nano-bubbles¹⁶ and the K-Q exciton in the heterostructure, we performed PL mapping on $WS_2/MoSe_2$ heterobilayer, which contains several bubbles (indicated by the arrow in Supplementary Fig. S13a,b) at the interface. We found that on top of the nanobubble, the WS_2 inlayer peak underwent a red shift due to strain, while the $MoSe_2$ peak remained unaffected (Supplementary Fig. S13c). Conversely, on the flat surface, we observed a prominent and highly intense S_Q peak. Additionally, the relative intensity of the bottom $MoSe_2$ layer increased at the flat surface, indicating strong coupling between the layers, which is deemed necessary for a high S_Q peak. Supplementary Fig. S13d,e, represents the deconvolution of each spectrum from the flat and nanobubble region, confirming the presence of a strong S_Q peak on the bubble-free flat surface. This finding again supports the conclusion that the S_Q peak is not originated from an interfacial bubble but rather arises due to strong interlayer coupling between the layers.

Fig. S13 | Effect of nano-bubble. **a**, Optical image of $WS_2/MoSe_2$ heterobilayer. Nano-bubbles are indicated by arrows. **b**, Spatial PL map of $WS_2/MoSe_2$ heterobilayer, revealing that bright spots are localized in spatially discrete regions corresponding to the presence of nano-bubbles. **c**, Comparison of PL spectra collected from flat (labeled as A: orange) and nanobubble region (labeled as B: blue). **d,e**, Deconvolution of the spectrum from the flat and nanobubble region.

Reviewer #3:

General comment:

In their manuscript entitled “Sequential order dependent dark-exciton modulation in bi-layered TMD heterostructures”, the authors claim that the optical properties of TMD hetero- and homo-structures depends in the stacking order of the individual layers. It results in the emergence of a peak in the photoluminescence at a lower energy than the A-exciton of the top layer that the authors assigned to be an intralayer indirect exciton. The authors have performed a thorough set of experiments on various samples. In this sense, the observation of stacking dependent optical properties is convincing. The manuscript is well written and referenced, and a lot of additional data support their observation. However, given that the interpretation of the effect is mainly attributed to strain, which is known to induce direct to indirect transition of the optical bandgap in TMDs, the novelty of this work seems limited. Their result could potentially be worth publishing in Nature Communication provided that the authors clarify the novelty of their finding as well as a few points of concerns:

Response: We are happy to hear that the manuscript is well-written, and referenced, and lot of additional data to support the main findings. We admit that the original manuscript lacks the proper explanation regarding the novelty of the work. The entire manuscript was revised to strengthen the introduction by focusing on the novelty of the work. We thank the reviewer for his/her valuable comments that allowed us to significantly improve the revised manuscript. For this, we would like to thank the reviewer for bringing all important comments. Please see the point-by-point responses below.

Comment 1: Previous works have demonstrated the that strain can induce drastic change of the optical bandgap and thus PL response e.g.[Nano Lett. 13, 3626-3630 (2013)] and can be used to engineer the optical properties of TMDs. In this sense, the work presented by the authors seems like a direct application of this concept. Can the author clarify the novelty of their work with respect to prior art on strain-based bandgap engineering?

Response: We appreciate these valuable comments. In terms of strain engineering, it holds true that the optical band gap can be efficiently modulated via the application of external strain [Nano Lett. 13, 3626-3630 (2013); Nano Lett. 13, 2931-2936 (2013); Nano Lett. 17, 5634-5640 (2017); Nature Communications 11, 1151 (2020); Adv. Mater. 2107362 (2022)]. However, **none of these previous studies observed the appearance of any new/additional PL peak with an application of strain, only peaks are shifted to blue or red depending on compressive or tensile strain respectively.** In the suggested reference paper [Nano Lett. 13, 3626-3630 (2013)] authors claim that a larger than 1% strain changes the lowest-lying bandgap from direct to indirect. However, there is no direct signature (PL peak) corresponding to indirect transition. Furthermore, **our result clearly demonstrated that even without the application of external strain on the system we can successfully modulate the optical properties of the material by changing the stacking sequence between the layer.**

Nevertheless, our main goals are completely different. This study represents a substantial invention as we have demonstrated, for the first time, that **altering the stacking sequential order in a heterostructure can lead to changes in its physical properties.** This unexpected

phenomenon challenged the traditional assumption that the intrinsic band structure remains unchanged by changing the stacking sequence. Our investigation revealed **the emergence or disappearance of additional photoluminescence peaks (S_Q/Seq) depending on the stacking sequence**, which has not been reported previously to date. We confirmed the reproducibility of these additional peaks in multiple heterostructures and explored various scenarios to explore such an unusual phenomenon. Finally, our microscopic many-particle model concluded that layer-dependent strain could be a possible solution and showed good agreement with experimental results. Furthermore, this newly observed stacking sequence-dependent effect requires further exploration, as we believe it has great potential to open up a new field of study known as **fliptronics**.

We like to summarize the novelty of our work in three aspects, such as – i) scientific advancement, ii) technological development, and iii) impact on the community.

i) Scientific advancement:

- (1) We newly observed momentum-forbidden **intra-layer K-Q exciton** in the TMD heterobilayer.
- (2) We first time demonstrated the **stacking sequence-dependent unusual dark exciton emission in bi-layered TMDs heterostructure**, which is distinct from the conventional insight that the optical properties remain unchanged with altering the stacking sequence in 2D heterostructure.
- (3) We introduce a **metric (σ) to identify the coupling strength** between the layers in the heterostructures region based on the K-Q PL intensity.

ii) Technological development:

- (1) This sequence-dependent study offers another **degree of freedom to choose the proper stacking sequence** on the same materias heterostructure to get **new optical properties**.
- (2) The PL intensity ratio of Q- to K-excitons in the same layer is inversely proportional to laser power, unlike for conventional K-K excitons. This can be utilized as **optical power switches** in solar panels.

iii) Impact on community:

- (1) The observed novel phenomena, stacking sequence-dependent anomalous optical properties, could **open a new field of study**, named ‘**fliptronics**’, similar to twistrionics.
- (2) This study proposes that unusual stacking sequences could be useful for further applications to engineer the optical properties of TMD heterostructures.

In summary, our approach attracts a broad readership in various fields like condensed matter physics, photonics, material science, and optoelectronic engineering.

Comment 2:

Part-I: The authors do not show, or I missed it, the PL from interlayer excitons (ILX) in addition to their S_Q and S_{eQ} signals. I understand this is not the scope of this work, but that would remove any ambiguity on the possibility that the observed peaks are originating from an interlayer excitonic state. In particular, the authors claim to have tested a large number of samples that are randomly stacked. One would expect to see data exhibiting both the additional peak they observe and the signature (or not) of an ILX depending on twist angle.

Response: Thank you for the comment on the interlayer exciton. We like to mention that our new peaks S_Q or S_{eQ} are not related to ILX, for example, as we mentioned on page 3, line 76 results in the original manuscript. Nevertheless, we agree with the reviewer, to remove any ambiguity, we need to explain more elaborately at this point. In our case, we have observed only a few times the signature of the interlayer exciton (ILE). For example, in $\text{MoSe}_2/\text{WSe}_2$ heterostructure, a weak ILE peak was observed at energy 1.345 eV at room temperature (Fig. R3-1), which is well matched with the previously reported results [Nature Communications 2015, 6, 6242; Phys. Status Solidi B 2019, 256, 1900308; Nature Communications 2021,12,1656]. However, in most of the cases, we have not observed any signature of interlayer excitons. The reason behind this is listed below -

1. In general, interlayer exciton requires specific angle alignment between the layer as observed previously [Nature 2019, 567, 76; Nature 2022, 610, 478; Nature Materials 2020, 19, 617; Nature Communications 2020, 11, 5888; ACS Nano 2017, 12, 4041]. However, in our case, heterostructures are not fabricated with angle alignment.

2. Interlayer excitons are most prominent at low temperatures [Phys. Status Solidi B 2019, 256, 1900308; Nature 2019, 567, 76; Nature 2022, 610, 478; Small 2019, 15, 1902424; Nature Materials 2020, 19, 617; Optics Express 2020, 28, 13260], nevertheless, in this work all the experiments have been conducted at room temperature unless specified.

3. Interlayer exciton generally appears, i.e., 1.35-1.45 eV as reported previously [Nature 2019, 567, 76; Nature 2022, 610, 478; Nature Materials 2020, 19, 617; Optics Express 2020, 28, 13260], which is beyond the scope of this work.

We believe that because of all these reasons stated above, the interlayer exciton does not appear in our case, except few of the samples, one of them shown below.

Figure R3-1 (Supplementary Fig. S10) | Interlayer exciton. An interlayer exciton (ILE) peak appears at energy 1.345 eV in MoSe₂/WSe₂ heterostructure at room temperature.

In the revised version of the supplementary, we have added a new figure (Supplementary Fig. S10) as well as a new paragraph to discriminate the interlayer exciton from our additional new PL peaks (S_Q/Se_Q).

Modifications to the manuscript (page 3, line 79):

Furthermore, the energies of these additional peaks (1.89 and 1.56 eV) are distinct from reported interlayer excitons (1.35-1.42 eV) for the WS₂-WSe₂ heterostructure.^{8,17} Moreover, we observed weak interlayer exciton peak at an energy of 1.35 eV in MoSe₂/WSe₂ heterostructure at room temperature (Supplementary Fig. 10). This is well matched with previously reported results,^{10,12,19} which is rarely observed as it is contingent upon a specific stacking angle.

Modifications to the Supplementary (page 10, line 154):

Note 6. Interlayer exciton

We rarely observed the interlayer exciton (ILE). For example, in MoSe₂/WSe₂ heterostructure, a weak ILE peak was observed at an energy of 1.35 eV at room temperature (Supplementary Fig. S10), which is well matched with previously reported results.⁴⁻⁶ However, in most of the cases, we have not observed any signature of interlayer excitons. The reason behind this is listed below:

- i) In general, interlayer exciton requires specific angle alignment between the layers as observed previously.⁷⁻¹¹ However, in our case, heterostructures are not fabricated with specific angle alignment.

ii) Interlayer excitons are most prominent at low temperatures.^{5,7-9,12,13} Nevertheless, in this work, all the experiments have been conducted at room temperature.

iii) Interlayer exciton generally appears, i.e., at the energy of 1.35-1.45 eV for WSe₂-WS₂ heterostructure as reported previously,^{7-9,13} which is beyond the scope of this work.

Fig. S10 | Interlayer exciton. Interlayer exciton (ILE) peak appears at an energy of 1.35 eV in MoSe₂/WSe₂ heterostructure at room temperature.

Part-II: How random is the stacking? Have the authors measured the twist angle of their samples?

Response: We appreciate the reviewer's query regarding the stacking angle. Considering this comment, we performed second harmonic generation (SHG) measurements on seven samples to determine the stacking angle between the layers (Table R3-1), where we observed the indirect K-Q peak. One of those has been illustrated in Fig. R3-2. Based on this observation, we may conclude that the appearance of our newly observed K-Q intralayer peak is independent of the stacking angle, making it more robust compared to moiré or interlayer excitons.

This demand that our observed intralayer K-Q exciton has great potential for future optoelectronic applications, as it is indirect and independent of the stacking angle and more importantly, PL intensity similar to monolayer intensity at room temperature. We thank the reviewer for this suggestion, it impacts greatly improves our manuscript.

Figure R3-2 (Supplementary Fig. S9) | Twist-angle measurements using second-harmonic generation (SHG). **a**, Optical image of WS₂/WSe₂ heterobilayer. **b,c**, Polarized SHG intensity polar diagram as a function of incident field polarization angle for WS₂ and WSe₂, respectively. The black dots correspond to experimental data and the red line is the fitting, which reveals the stacking angle between these two-layer is -7.09°.

Table R3-1 (Supplementary Table S1) | Stacking angle measured via second harmonic generation (SHG).

Sample name	Stacking sequence (top/bottom) layer	Stacking angle (degree)
RD13	WS ₂ /WSe ₂	16.36
RD14	WSe ₂ /WS ₂	15.14
RD26	WS ₂ /WSe ₂	8.48
RD51	WS ₂ /WSe ₂	24.53
RD52	WS ₂ /WSe ₂	24.31
RD64	WS ₂ /WSe ₂	-7.09
RD73	WS ₂ /WSe ₂	35.34

Modifications to the manuscript (page 3, line 73):

We note that all heterostructures were randomly stacked and the stacking angle between the layers was further confirmed by second harmonic generation (SHG) measurements (Supplementary Fig. 9). Therefore, the observed features are not specific to a stacking angle.

Modifications to the Supplementary (page 9, line 139):

Note 5. Stacking angle measurements

We performed second harmonic generation (SHG) measurements on seven different samples to determine the stacking angle between the layers (Table S1), where we observed the indirect K-Q peaks, as illustrated in Supplementary Fig. 9. Based on the observations, we conclude that our newly observed K-Q intralayer peak is independent of the stacking angle, making it more robust compared to moiré or interlayer excitons.

Fig. S9 | Twist-angle measurements using second-harmonic generation (SHG). **a**, Optical image of WS₂/WSe₂ heterobilayer **b,c**, Polarized SHG intensity polar diagram as a function of incident field polarization angle for WS₂ and WSe₂ respectively. The black dots correspond to experimental data and the red line is the fitting, which reveals the stacking angle between two layers is -7.09° .

Table S1 | Stacking angle measured via second harmonic generation (SHG).

Sample name	Stacking sequence (top/bottom) layer	Stacking angle (degree)
RD13	WS ₂ /WSe ₂	16.36
RD14	WSe ₂ /WS ₂	15.14
RD26	WS ₂ /WSe ₂	8.48
RD51	WS ₂ /WSe ₂	24.53
RD52	WS ₂ /WSe ₂	24.31
RD64	WS ₂ /WSe ₂	-7.09
RD73	WS ₂ /WSe ₂	35.34

Part-III: Since the samples are heated up during the fabrication process this could favor relaxation of the top layer along some preferential orientations.

Response: Relaxation of the top layer to have preferential orientation is very unlikely in our case even after heat treatment. While the top layer is placed to the bottom layer they are immediately coupled by Coulombic force. Moreover, the thermal energy actually helps to improve the coupling strength between the layers, which further ensures the tight adhesion between the layers [2D mater, 2018, 5, 031001; 2D mater, 2017, 4, 021019].

Comment 3:

Part-I: I am not sure I fully understand how that the authors can claim that the peak from the top layer originates from K-Q excitons. I understand that in TMD HS, K valleys of the two layers do not couple and that coupling between bands of the layers varies to be the strongest at Gamma. Is that why the Q-band is renormalized and not K?

Response: We understand that two layers of highly stacking order, like AA' may involve momentum coupling between layers. However, in our case, all hetero-bilayers are randomly stacking ordered as explained by the previous comment (Supplementary Fig. S9), the coupling strength between two layers is exclusively dependent on van der Waals interaction. More importantly, the Q-bands are more sensitive and change drastically than K-band. One reason could be as the reviewer suggested K valleys of two layers do not strongly couple in TMD heterostructures. Moreover, *Thorsten et al* in their article also claim that Q-band is very sensitive to the lattice constant rather than K-band [2D Matter. 6, 035003, (2019)]. In our scenario, since the bottom layer is directly exfoliated on the substrate, and it freezes with the substrate, whereas the top layer is free, so it adjusts the lattice constant according to the bottom layer. Thus Q-band is drastically renormalized at the top layer. Further, the DFT calculation from recently published reports also confirms that not K-K (which was commonly assumed to be the ground state), but rather K-Q is the ground state interlayer exciton [Nature Materials, 19, 617-623, (2020); Phys. Rev. B, 97, 165306, (2018)], as the Q-band drastically changes at the heterostructure and energetically resided down compared to K-band depending on the material. Of course, it always depends on the position of the Q-band, if it is too far from the K-band then Q-band is downshifted however it cannot cross the K-band, for example, in Mo-based materials [2D Matter. 6, 035003, (2019)] and therefore, it gives rise to energetically higher K-Q peak (Fig. 1f) compared to K-K. To establish this, we have calculated the excitonic band dispersion for K-Q excitons for (a) WS₂ and (b) MoSe₂ monolayers with and without strain as shown in Fig. R3-3 (red and blue, respectively). The excitonic energies are microscopically determined by numerically solving the Wannier equation. Comparing the energy of K-Q excitons to the K-K exciton energy E_{K-K} , we see that K-Q excitons are energetically below and above K-K for (a) WS₂ and (b) MoSe₂, respectively - exactly as the peaks S_Q and Se_Q in Figs. 1e-f, respectively.

Figure R3-3 (Supplementary Fig. S27) |Dispersion of dark excitons. Energy of K-Q excitons as a function of the center-of-mass momentum for unstrained (blue) and compressively strained (red) **a**, WS₂ and **b**, MoSe₂.

In conclusion, both K- and Q-band are renormalized, but the Q-band is renormalized more significantly and resides at lower energy than K-band (W-based materials).

Part-II: Is there a possibility for the indirect exciton to be Gamma-Q or Gamma-K (with holes at Gamma)? **Response:** We thank the reviewer for the fascinating query regarding the possible choices of momentum indirect exciton including Γ -K or Γ -Q. There could be the possibility for indirect Γ -K or Γ -Q exciton in the bilayer, where holes reside at Γ in the valance band edge. However, the choice of the most accessible states depends on the band alignment at the heterostructure. If the energy of the Γ -band in the valance band is higher than that of K-band, the Γ -K or Γ -Q exciton would yield the prominent exciton. However, in our case, K-Q is the most probable exciton, and the reasons to support this are elaborated below:

- (i) Based on DFT calculation on recently published literature for all types of different stacking configurations for a variety of TMDs [*Nature Materials*, 19, 617-623, (2020); *Phys. Rev. B*, 97, 165306, (2018); *Phys. Status Solidi B*, 258, 2000614, (2021); *Appl. Phys. Lett.* 108, 063105, (2016); *Scientific Report*, 3, 1549, (2013); *2D Mater.* 6, 035003, (2019)], the K position always resides at higher energy in the valance band than any other bands. This confirms that the K-band in the valance band is the most preferable to any other momentum position in the Brillouin zone.
- (ii) The energy difference between K and Q bands (ΔE_{K-Q}) in the conduction band of WS_2 in WS_2/WSe_2 heterostructure is 88-62 meV from the previous study [*Nat. Mater.* 19, 617-623 (2020). This energy difference is consistent with the energy difference between the inlayer WS_2 peak and the S_Q peak observed in our experiment.
- (iii) *Thorsten Deilmann* et al. claimed that for Mo-based TMDs ($MoSe_2$ or MoS_2), the lowest energy excitation is the Γ -Q transition [*2D Mater.* 6, 035003 (2019)]. The transition energy of Γ -Q of MoS_2 or $MoSe_2$ in the bilayer was 1.84 eV or 1.70 eV. However, in our experiment, we did not observe such additional peaks in this energy range. On the other hand, for W-based materials (WS_2 , WSe_2) K-Q transactions are energetically lowest, as they found that vanishing oscillatory strength for Γ -Q, whereas non-zero for K-Q. This result again well matches our results for K-Q transition.

Due to these above reasons, we conclude that in our case K-Q is the most preferable transition rather than Γ -K or Γ -Q.

Part-III: If this renormalization is more pronounced for strongly coupled layers, as the authors show, why doesn't it affect the Q band of the bottom layer?

Response: We thank the reviewer for this important query. In the previous discussion (*part -I*) we understand that the Q-band is more sensitive than K-band. Now it is interesting to see that Q-band is more sensitive on the top layer. We have elaborated underlying reasons below–

- (i) According to our hypothesis, the bottom layer is directly exfoliated on the substrate and freezes with the substrate. Meanwhile, the top layer is relatively free to be relaxed (see *Fig. R3-4*). In the previous article *2D Mater.* 6, 035003, (2019), the

authors claim that Q-band is more sensitive to the change of lattice constant than the K-band. Therefore, there is a higher probability of affecting the Q-band in the top layer compared to the bottom.

Figure R3-4 | Strin on the top layer. Schematically represent that the bottom layer is fixed by the substrate whereas the top layer experience a compressive strain.

- (ii) To validate our hypothesis, we introduced a theoretical model (Elliot formula) that includes indirect phonon-assisted recombination of momentum-dark excitons. By applying a small compressive strain on the top layer (-0.3% for the top-WS₂ layer and -0.15% for the top-WSe₂ layer) in heterostructures, we were able to successfully reproduce the experimental data (Fig. 4e,f).
- (iii) Previously, the conduction band splitting in bilayer MoSe₂ has been estimated [Nat. Nanotech. 15, 750-754, 2020: Supplementary Fig. 6]. Authors claim the electrons are more populated in the top layer than in the bottom layer since the top layer resided at a lower by ~50 meV than the bottom layer at K-point (see Fig. R3-5). Furthermore, the Q-band was drastically downshifted only on the top layer, compared to the bottom layer.

Figure R3-5 | Conduction band spitting between the top and bottom layer MoSe₂ [Reference: Nat. Nanotech. 15, 750-754, 2020].

Considering this comment, we have added a new schematic in the supplementary material to clearly demonstrate the freezing of the bottom layer whereas the top layer changes the lattice parameters.

Modifications to the Supplementary (page 25, line 401):

Note 20. Schematic representation of compressive force

Fig. S26 | Strain on the top layer. Schematical representation that the bottom layer is fixed by the substrate whereas the top layer experiences a compressive strain.

Part-IV: The authors perform STS measurements from which they conclude that the low energy band is located in the top layer. If there is a strong coupling between layers, shouldn't one expect an overlap of the wavefunctions between the layers?

Response: Thank you to the reviewer for bringing up this important point again. We would like to mention that in this study, the heterostructure was prepared in a way to perform the STS on each monolayer region, as well as the heterostructure region (Fig. 2e). We observed an additional density of states (Fig. 2f: red color) near the conduction band edge exclusively at the heterostructure region. Therefore, as the reviewer asked, one can expect this additional density of states to emerge from the bottom layer due to the overlap of the wavefunctions between the layer. This is, however, not the case in STS measurements. When we measured the tunneling current between the W-tip and the material, the tunneling current was optimized to choose tip separation distance which remained fixed during measurements of monolayer or heterostructure. While the tunneling current was optimized on the top layer in the heterostructure, the bottom layer in the heterostructure is placed away by ~ 1 Å from the top layer, and the tunneling current decays exponentially with distance [C. Julian Chen. *Introduction to Scanning Tunneling Spectroscopy. Vol.15, 2016*]. Therefore, the tunneling can be negligible from the bottom layer.

The contribution from the bottom layer can also be measurable for the 'Close tip Condition' [Nat. Commun. 6, 7666, 2015; Nano Lett. 16, 4831-4837, 2016], where the tip is placed very close to the surface to achieve the contribution from the bottom layer as well. However, this has not been considered in this work.

Part-V: I am not sure I understand how they come to this conclusion and how they can separate contributions from the two layers.

Response: The contribution of the top and bottom layers has been identified in these ways –

- (i) **Fixing the top layer:** We fixed the top layer (say WS_2) and varied the bottom layer such as - (a) WSe_2 , (b) $MoSe_2$, (c) MoS_2 , and (d) WS_2 itself as shown in Fig. R3-6. Remarkably, the S_Q peak (red color peak) remains the same in all cases, except for a slight variation of the energy due to band renormalization with different TMD combinations. The presence of the S_Q peak in all cases confirms that it originates from the WS_2 layer, which was placed on top of all the heterostructure.

Figure R3-6 | Fixed top layer. S_Q always appears while fixing the top layer (WS_2).

- (ii) **Q-band of top layer:** In previous study [Nat. Nanotech. 15, 750-754, 2020: Supplementary Fig. 6], the conduction band splitting in bilayer $MoSe_2$ was estimated, and it was found that electrons are more populated in the top layer than the bottom layer. Further, the authors demonstrated that the Q-band was drastically downshifted only on the top layer. Our current observations are consistent with these results, where we found that the Q-band was significantly downshifted only in the top layer, as discussed in part II of question 3.

In this way, we confirm that this newly observed PL peak (K-Q peak) originates from the top layer of the heterostructure. This study further highlights the importance of the stacking sequence when designing 2D heterostructures.

Comment 4: If the change in the Q-band is related to strain, I am surprised not to see some similar signature in the bottom layer. I would be surprised that all the bottom layers of the studied samples are strain free. This brings me to the PL response seen in Fig. 1d, S8c, S15a, in which it is the clearest. In those measurements, the bottom layer shows a broader emission around the A-exciton than when it is on top. Could it be that the indirect exciton peak is also present for the bottom layer but with a much smaller energy separation from the A-exciton and a different relative intensity?

Response: We thank the reviewer for critically perceiving our data. Initially, we also thought the same that the bottom layer has the same K-Q peak with smaller intensity in the bottom WS₂ layer. However, it was not fit well with all data. Further,

- (i) In a particular laser power, PL yield is different for WS₂ and WSe₂ due to different intrinsic carrier concentrations. In the WS₂ case, trions emerge as low as 0.33 uW laser power confirmed by our previous study [See *ACS Nano*. 15, 2849-2857, 2021]. In his study (Fig. 1d) we use higher laser power (~50 μW), thus trion peaks overlap with neutral excitons with similar energy ($\Delta E \sim 27$ meV), well-fitted with trions and neutral peaks (inset of Fig. R3-7). Therefore, due to the contribution of trion along with neutral exciton PL peak become wider in this case.

Figure R3-7 | Trion contribution. PL spectra of WSe₂(top)/WS₂(bottom) heterobilayer. Wider WS₂ peak due to contribution of trion along with neutral exciton. Inset: peak deconvolution of the neutral exciton (olive) and trion (green) with an energy difference of 27 meV.

- (ii) In Fig. S15a, the PL peak broadens due to the same reason, that is a contribution of trion again at heterostructure as we used higher laser power compared to the monolayer region. Further from our previous study [*ACS Nano*. 15, 2849-2857, 2021], we investigated that Re- doping actually increases the intrinsic carrier concentration of WS₂ material. Thus, in this case, trion becomes more dominant compared to neutral exciton.

- (iii) Fig. S8c did not use the same power range as well as different devices. Thus peak broadness is different in these cases as well. In order to remove the confusion we have modified Fig. S8 and add the data for both stacking configurations.
- (iv) Nonetheless, if we consider the other figures, where the trion peak is not dominant (Fig. 1b, Extended data Fig. 2b, Supplementary Fig. S13a) with similar stacking configuration (WSe₂(top)/WS₂(bottom)), we observed in these cases the broadness of the peaks similar to monolayer.

In conclusion, due to *different laser powers*, we occasionally observed broad PL yield with multi-excitons such as trion and/or bi-exciton but is not related to K-Q excitons.

The reason behind the strain is only on the top layer, which we discussed in the previous part -III of question number 3.

Modifications to the Supplementary (page 9, line 130):

Note 4. Identification of the emerging dark exciton peak

Fig. S8 | Presence of Se_Q and S_Q peak. **a**, PL spectrum of monolayer WSe₂. **b**, PL spectrum at top-WSe₂/bottom-WS₂ heterostructure, where clear Se_Q emerges at an energy of 1.56 eV. **c**, PL spectrum at top-WS₂/bottom-WSe₂ heterostructure, where the Se_Q peak is absent. **d**, PL spectrum of monolayer WS₂. **e**, PL spectrum at the top-WSe₂/bottom-WS₂ heterostructure, where Se_Q is absent. **f**, PL spectrum at the top-WS₂/bottom-WSe₂ heterostructure, where the prominent S_Q peak is present.

Comment 5: As the authors claim, emission from indirect excitons has to be phonon-assisted. That process should be very sensitive to temperature. Have the authors performed a temperature dependence to see whether the photoluminescence is in agreement with previous work, e.g. [Brem et al., Nano Lett. 20, 2849-2856 (2020)]?

Response: We thank the reviewer for raising this important point. The phonon-assisted photoluminescence is indeed a strongly temperature-dependent process, as we have shown in our previous work mentioned by the reviewer [Nano Lett. 20, 2849 (2020)]. While the theoretical results presented in the main paper are taken at 77 K, we expect a smaller ratio between the bright exciton and the S/Se_Q peak at higher temperatures. Nevertheless, compared to the case of the unstrained hBN-encapsulated WSe₂ monolayers [Nano Lett. 20, 2849 (2020)], the visible phonon sidebands are expected to appear also at higher temperatures thanks to a larger dark-bright exciton separation (of almost 87 meV for the case of WS₂ under 0.3% compressive strain) and thanks to the 3-fold degeneracy of the emitting K-Q excitons.

In the revised manuscript we have added a new paragraph discussing the role of temperature for the PL.

Modifications to the Manuscript (page 17, line 460):

In particular, the strain in the top layer induces new peaks in good agreement with the experimental observation. The phonon-assisted emission has a lower efficiency than direct recombination, which is, however, compensated by a higher occupation of the K-Q excitons compared to the bright excitons. The higher relative occupation of K-Q excitons is induced by the energy alignment in tungsten-based materials and is further triggered by compressive strain. Furthermore, at decreased temperatures (77 K), we find an increased relative occupation of K-Q excitons and hence more intense phonon-sidebands.³¹ In contrast, in Mo-based materials, K-Q excitons are higher in energy than the bright-excitons even in the presence of compressive strain, Supplementary Fig. 27. This leads to a negligible phonon-assisted PL. Nevertheless, the strain-induced reduction of the dark-bright exciton separation (Supplementary Fig. 28) could potentially lead to a resonance similar to Se_Q in MoSe₂/WS₂, if additional activation mechanisms are present, e.g., via the interplay of strain and defects.³⁷ The microscopic evaluation of these mechanisms goes beyond the scope of this work.

Comment 6: In their final discussions, the authors suggest to use their finding for optical switch, which is an interesting approach. Have the authors considered other potential applications, such as valleytronics. Would it be possible to valley-polarize those indirect excitons, and take advantage of their long lifetime (with respect to the direct exciton)?

Response: Thank you for this query. We demonstrate the optical switch with an intensity ratio of K-Q to K-K exciton as an application. We did not show further applications as suggested for Valleytronics. This certainly opens new possibilities for potential applications.

Modifications to the Manuscript (page 9, line 245):

However, experimentally observed dark excitons of TMD heterostructures with inevitable use of substrate can be engineered for further exciton dynamics and can be useful as an optical power switch. **In addition, intralayer indirect excitons in heterostructures may hold significant potential for future valleytronics applications.**

Reviewers' Comments:

Reviewer #1:

Remarks to the Author:

The revised manuscript by Riya Sebait et al have given a very detailed point to point response and modification. In general, I agree with most of the explanations given by the authors. Moreover, the author has added a lot of experiments. On the current basis, in my opinion, the manuscript can be published after following minor revisions.

1. In the supplemental absorption data, I do not see any peaks that corresponding to K-Q dark exciton. Why?
2. I still believe that time-resolved PL characteristic is key evidence for the existence of K-Q excitons.

Reviewer #2:

Remarks to the Author:

In the revised manuscript, the authors provide new experimental evidences and more detailed interpretations to clarify some of my comments. I still have several concerns on how they interpret their PL spectra. It is well known that monolayer TMDs have very complex exciton emission states, e.g. neutral, trion, biexciton states and localized states. The authors need to be very careful to assign these peaks. For example,

1. In the new Supplementary Fig. S13, the authors fabricated new samples with new PL measurements to clarify the effect of nanobubbles. In Fig. S13d and Fig. S13e, they deconvolute PL peaks and assign two of high energy peaks to WS₂ neutral exciton and trions. The trion binding energy in monolayer TMDs has been well determined to be around 30 meV from many previous measurements (Nature Materials 2013, 12, 207–211; Nature Communications 2016, 7, 12715). And also, as shown in the Figure R2-5 in the response letter, the trion exciton emission is ~ 30 meV below the neutral exciton emission. However, when we look at Fig. S13d and Fig. S13e, the energy difference between assigned neutral exciton and trion is huge (~ 100 meV in Fig. S13e). The authors need to provide more interpretations or evidences to support their peak assignments and they are still not ruling out the possibility of localized excitons.

2. In the new Figure R2-3b as shown in the response letter, the authors assigned the lowest energy tail to the SeQ peak. How they rule out the possibility of localized exciton and biexciton?

Reviewer #3:

Remarks to the Author:

The authors has provided a very thorough and extensive reply. They have provided additional data to clarify their interpretation that the observed signal in PL originates from an indirect exciton. They clearly put a lot of effort in their reply and revised manuscript which significantly improves its clarity with a better presentation of the novelty from their finding. Therefore, I think their revised manuscript is worth publishing in Nature Communications.

REVIEWER COMMENTS

Reviewer #1 (Remarks to the Author):

General comment:

The revised manuscript by Riya Sebait et al have given a very detailed point to point response and modification. In general, I agree with most of the explanations given by the authors. Moreover, the author has added a lot of experiments. On the current basis, in my opinion, the manuscript can be published after following minor revisions.

Response: We are delighted to receive the comments from the reviewer that our response letter provided very detailed point-to-point responses. We sincerely thank the reviewer for appreciating our efforts and agreeing with our explanations. We greatly appreciate the reviewer for carefully assessing both our manuscript and response letter, and consider our manuscript deemed worthy of publication after minor revisions. Considering all the valuable comments and suggestions raised by the reviewer, we have addressed all the queries in a pointwise manner in the following sections.

Comment 1: In the supplemental absorption data, I do not see any peaks that corresponding to K-Q dark exciton. Why?

Response: We appreciate the reviewer for bringing up this query. The absence of K-Q peak in absorption measurement can be attributed to its indirect nature or momentum mismatch, as demonstrated in previous studies [ACS Nano 7, 791-797 (2013); Nanomaterials 8, 725 (2018); Nanotechnology 27, 115705 (2016)]. Such phenomena can be exemplified by comparing the layer-dependent PL and absorption measurements simultaneously. It is well known that as the number of layers increases, the band gap undergoes a transition from direct to indirect. Consequently, an additional indirect PL peak emerges with an increasing number of layers (due to not much energy difference between direct and indirect energy) [ACS Nano 7, 791-797 (2013); Nanotechnology 24, 465705 (2013)]. However, this indirect peak is absent in absorption measurement [Nanotechnology 27, 115705 (2016); ACS Nano 7, 791-797 (2013); Nanomaterials 8, 725 (2018)]. For example, Zhao et al [ACS Nano 7, 791-797 (2013)] demonstrated the appearance of an indirect PL peak starting from the bilayer, which further redshifted with an increasing number of layers for WS₂ and WSe₂ (red dotted lines Fig. R1-1 a,b). Meanwhile, for the same samples, no additional peak was observed in absorption measurement due to momentum mismatch (Fig. R1-1 c,d). To illustrate the similar behavior, we also measured the absorption for WS₂ mono- and bi-layer and directly compared it with the corresponding PL spectra (Fig. R1-1 e,f).

Figure R1-1 | Absorption measurement. **a,b**, Layer-dependent PL spectra for WS₂ and WSe₂ respectively, Peak I indicates an indirect gap emission, which emerges from the bilayer and was further red-shifted with the number of layers. **c,d**, Layer-dependent absorption spectra of WS₂ and WSe₂, where absorption peaks are redshifted with an increment of the number of layers [ACS Nano 7, 791-797 (2013)]. **e,f**, PL, and absorption spectra of mono and bi-layer WS₂.

Nevertheless, even without the emergence of an additional peak due to an indirect bandgap, the material absorption peak undergoes a redshift and broadening as the number of layers increases. This behavior has been attributed to the transition from direct to indirect bandgap as reported by several research groups [Scientific Report 5, 9218 (2015); Nanomaterials 8, 725 (2018); Journal of Optical Society of America B 35, 1183 (2018); Nanotechnology 27, 115705 (2016)]. In our case, we also observed similar results, where the bilayer WS₂ peak is redshifted and broadened compared to the monolayer (Fig. R1-1 e,f). This finding is consistent with WS₂/MoS₂ and WS₂/WSe₂ heterostructures compared to each monolayer (Supplementary Fig S19), which is analogous to the redshift observed in bilayer compared to monolayer [ACS Nano 7, 791-797

(2013); *Nanomaterials* 8, 725 (2018); *Nanotechnology* 27, 115705 (2016)]. This behavior has been attributed to the transition from direct to indirect bandgap (described in Supplementary Note 13).

Comment 2: I still believe that time-resolved PL characteristic is key evidence for the existence of K-Q excitons.

Response: We would like to thank the reviewer again for this suggestion, and we also agree with the reviewer that the time-resolved photoluminescence (TRPL) characteristic could be provided for additional evidence. We have measured the TRPL for our samples both in monolayer as well as on heterostructure region as shown in Fig R1-2a. The grey shadow area in the figure corresponds to the instrument response function (IRF). We observed two relaxation times for each individual material and three relaxation times at heterostructure, indicating a complex behavior with multiple relaxation processes occurring simultaneously. For each individual material, we found fast relaxation times (τ_1) of approximately 0.974 ns for WS₂ and 1.406 ns for WSe₂, attributed to exciton scattering or electron-phonon scattering [*ACS Nano* 14, 15374 (2020)]. Additionally, slow relaxation times (τ_2) of approximately 2.624 ns for WS₂ and 3.326 ns for WSe₂ were observed, likely due to trions and/or traps [*npj 2D Mater. Appl.* 2, 30 (2018)]. Both WSe₂ and WS₂ monolayers exhibited similar behaviors in terms of their relaxation times. The longer PL relaxation times observed for WSe₂ compared to WS₂ could be related to a higher defect density in WSe₂, as the presence of defects can introduce additional trap states and impurities that can influence the exciton dynamics and contribute to a longer lifetime. Notably, at the heterostructure, additional faster relaxation (~0.587 ns) appears, close to the range of IRF. This is consistent with the previous report on faster relaxation time in multilayered WS₂ compared to monolayer as shown in Fig R1-2b,c. This reduction of relaxation time in heterostructure can be attributed to fast charge transfer facilitated by both direct K-K band and indirect K-Q band transition, which primarily depends on strong interlayer coupling, similar to previous reports [*Nanoscale* 7, 7402 (2015); *ACS Nano* 14, 15374 (2020)].

Figure R1-2 | TRPL measurement. **a**, TRPL measurement of monolayer WSe₂, WS₂ and WS₂-WSe₂ heterostructure (our data). **b,c**, Layer-dependent PL and TRPL spectra for WS₂. Peak I indicates an indirect gap emission, which emerges from the bilayer and was further red-shifted with the number of layers [Nanoscale 7, 7402 (2015)].

In this report, we focused solely on the K-K and K-Q excitons in the heterostructure. To accomplish this, we need to perform simultaneous measurements of each K-K and K-Q peak and their corresponding relaxation time. However, simultaneous measurements of such peaks cannot be possible, primarily due to their close proximity (≥ 50 nm wavelength). As a result, performing such precise measurements are beyond the scope of the current work.

Modifications to the Manuscript (page 5, line 135):

The momentum indirect phonon-assisted emission from intralayer K-Q exciton becomes stronger by accumulated electron population at the Q-band transferred from the K-band in the same WS₂ layer as well as additional electron population transferred from WSe₂ to WS₂ layer due to type-II band alignment in the heterostructure (Supplementary Fig. 20a,b).^{8,28,29} **Additionally, the reduction of relaxation time in heterostructure, measured by time-resolved photoluminescence, can be attributed to fast charge transfer facilitated by both direct K-K band and further indirect K-Q band transition (Supplementary Fig. 20c).** This gives rise to strong emission of the S_Q peak.

Modifications to the Supplementary (page 21, line 331):

Note 14. Charge transfer and the related exciton dynamics

The charge transfer at the heterostructure is the key to explain the intensity modulation of K and Q-excitons. The PL intensity of neutral excitons of WSe₂ at heterostructure was greatly reduced compared to the individual WSe₂ monolayer region (Supplementary Fig. S20a) due to efficient charge transfer from WSe₂ to WS₂. Meanwhile, Q-exciton intensity was remarkably developed, whereas the intensity of WS₂ K-exciton was also reduced at the heterostructure due to further charge transfer from K to Q-band (Supplementary Fig. S20b). Supplementary Fig. S20c shows two relaxation times of fast relaxation τ_1 (WS₂: ~0.974 ns and WSe₂: ~1.406 ns) due to presumably exciton scattering or electron-phonon scattering²⁴ and slow relaxation time τ_2 (WS₂: ~2.624 ns and WSe₂: ~3.326 ns) due to trions and/or traps.²⁵ Such behaviors of two relaxation times of individual WSe₂ and WS₂ monolayers are similar to each other. Notably, additional faster relaxation (~0.587 ns) appears at the heterostructure. This can be explained again by the efficient charge transfer from WSe₂ to WS₂ and further from K to Q-band in WS₂. This is consistent with the previous report on faster relaxation time in multilayered WS₂ compared to monolayer.²⁶

In this report, we focused solely on the K-K and K-Q excitons in the heterostructure. To accomplish this, we need to perform simultaneous measurements of each K-K and K-Q peak separately and their corresponding relaxation time. However, simultaneous measurements of such discrete peaks cannot be possible, primarily due to their close proximity (as wavelength less equal to 50 nm). As a result, performing such precise measurements are beyond the scope of the current work and requires further investigations.

Fig. S20 | Charge transfer at heterostructure. a,b, PL measured at individual monolayers (WSe₂: blue-dotted line; WS₂: green-dotted line) as well as WS₂/WSe₂ heterostructure (yellow solid line) at the same excitation power. PL intensity is drastically reduced on WSe₂ side compared to WS₂ at the heterostructure region, which indicates the majority of the charge transferred from WSe₂ to K and Q-band of WS₂ as represented in **b**. **c,** TRPL measurement of monolayer WSe₂, WS₂ and WS₂/WSe₂ heterostructure.

Reviewer #2 (Remarks to the Author):

General comment:

In the revised manuscript, the authors provide new experimental evidences and more detailed interpretations to clarify some of my comments. I still have several concerns on how they interpret their PL spectra. It is well known that monolayer TMDs have very complex exciton emission states, e.g. neutral, trion, biexciton states and localized states. The authors need to be very careful to assign these peaks.

Response: We would like to express our gratitude to the reviewer for the insightful suggestions. We appreciate the reviewer's acknowledgment of our efforts. We are truly grateful for the reviewer's dedication to this work for carefully and critically assessing both our manuscript and response letter. We conducted a thorough analysis and carefully rechecked each of the assigned peaks as mentioned by the reviewer. Please find our point-by-point responses below.

Comment 1: In the new Supplementary Fig. S13, the authors fabricated new samples with new PL measurements to clarify the effect of nanobubbles. In Fig. S13d and Fig. S13e, they deconvolute PL peaks and assign two of high energy peaks to WS₂ neutral exciton and trions. The trion binding energy in monolayer TMDs has been well determined to be around 30 meV from many previous measurements (Nature Materials 2013, 12, 207–211; Nature Communications 2016, 7, 12715). And also, as shown in the Figure R2-5 in the response letter, the trion exciton emission is ~ 30 meV below the neutral exciton emission. However, when we look at Fig. S13d and Fig. S13e, the energy difference between assigned neutral exciton and trion is huge (~ 100 meV in Fig. S13e). The authors need to provide more interpretations or evidences to support their peak assignments and they are still not ruling out the possibility of localized excitons.

Response: We sincerely thank the reviewer for bringing up these important points. We agree with the reviewer regarding the energy difference between trion and neutral exciton, which is approximately ~30 meV. In light of this, we conducted a thorough analysis and carefully reexamined our data. We have now included the revised figure (see Fig. R2-1) in the new version of the supplementary information (Supplementary Fig. S13 d,e). Based on our current analysis, we observed that both flat (clean) and bubble heterostructures exhibit a neutral exciton (~ 2.00 eV) and trion peak (~1.96 eV) separated by approximately ~ 40 meV apart, which is consistent with previous reports [Nature Communications 7, 12715 (2016); ACS Nano 15, 2849 (2021); Small 14, 1703727 (2018); Nature Materials 12, 207–211 (2013); Scientific Report 5, 9218 (2015)]. Indeed, the energy separation between excitons and trions is influenced by intrinsic carrier modulation, which can be achieved either optically or electrically [Nature Materials 12, 207–211 (2013); Scientific Report 5, 9218 (2015); ACS Nano 15, 2849 (2021)]. Additionally, the inter-layer distance between vdW heterostructures also plays a crucial role in modulating this energy separation [Nano Letters 18, 2725 (2018)].

Figure R2-1 (Supplementary Fig. S13) | Effect of nano-bubble. **a**, Optical image of WS₂/MoSe₂ heterobilayer. Nano-bubbles are indicated by arrows. **b**, Spatial PL map of WS₂/MoSe₂ heterobilayer, revealing that bright spots are localized in spatially discrete regions that correspond to the presence of nano-bubbles. **c**, Comparison of PL spectra collected from a flat (labeled as A: orange) and a nan-bubble region (labeled as B: blue). **d,e**, Deconvolution of the spectrum from the flat and nano-bubble region.

In our observations, we identified an additional peak at an energy of approximately 1.91 eV in the nanobubble region. This peak exhibited similarities to localized excitons observed in a previous study [Nature Nanotechnology, 15, 854–860 (2020)]. However, it is a challenging task to differentiate between the localized excitons and the contribution of K-Q excitons, since their energy ranges overlap only at the bubble region. A similar situation was observed in the intact bilayer case, where the indirect peak (due to the bilayer) could overlap with its localized exciton [ACS Nano 7, 791-797 (2013); Nanoscale 7, 7402 (2015); Nature Nanotechnology, 15, 854–860 (2020)].

Nevertheless, the presence of a strong S_Q peak in the flat clean heterostructure region rules out the possibility of localized excitons. Because in previous reports have shown that at flat and clean surfaces, no additional peaks are observed (see Fig. R2-2). Moreover, the relative intensity of the bottom MoSe₂ layer increased at the flat region, indicating strong coupling between the layers, which is deemed necessary for a high S_Q peak.

Figure R2-2 | Previous report about the localized exciton from nano-bubbles [Reference: Nature Nanotechnology, 2020, 15, 854–860]. **a**, Comparison of PL spectra collected from a flat and nano-bubble region. **b**, AFM topographic image of WSe₂/h-BN heterostructure. **c**, Special map of the localized exciton.

Additionally, it has been well established by several groups [2D mater, 2018, 5, 031001; 2D mater, 2017, 4, 021019] that *post-annealing effectively removed nanobubbles (as well as localized exciton)*. In our case, we observed K-Q remained intact before and after annealing as shown in Supplementary Fig. S17. In conclusion, although distinguishing between localized excitons and K-Q excitons is challenging due to energy overlap in the presence of nanobubbles, however, the existence of a strong S_Q peak in the flat bubble-free region confirms that this S_Q peak is not related to nanobubbles.

Once again, we express our gratitude to the reviewer for providing this valuable suggestion, which has allowed us to further enhance the clarity of our findings.

Comment 2: In the new Figure R2-3b as shown in the response letter, the authors assigned the lowest energy tail to the Se_Q peak. How they rule out the possibility of localized exciton and biexciton?

Response: We appreciate the reviewer’s concern. We will now address each point individually to demonstrate how we have ruled out the possibility of localized exciton and biexciton.

Biexciton:

- i. **Energy difference:** The energy difference between neutral exciton to biexciton is two times the energy difference between neutral to trion, which does not fit well with our data. From the deconvolution of mentioned Fig. R2-3b, (shown in Fig. 1b in the main manuscript), the energy difference between neutral exciton to trion ~ 20 meV and neutral exciton to Se_Q peak ~ 80 meV. This difference in energy supports that the assigned peak is not related to biexciton.
- ii. **Laser power:** The presence of S_Q/Se_Q peak even at very low laser power (~ 3.5 uW) indicates that it does not depend on laser power, often associated with the formation of higher excitonic states like biexciton.

- iii. **Room temperature:** Bi-exciton feature is more prominent at low temperatures and makes it challenging to distinguish at room temperature. However, in our case, we observed a very distinguished and prominent peak (in some cases even it is higher than neutral exciton depending on coupling strength) at room temperature. This behavior suggests that the additional peak corresponds to a different excitonic state, rather than a biexciton.
- iv. **Power law:** Biexciton can be identified by the quadratic relation between population counts of biexciton (I_b) and neutral exciton (I_n), represented as $I_b \propto I_n^\alpha$, where α should be 2 for ideal bi-exciton [Nat. Phys. 11, 477 (2015); ACS Nano 9, 647 (2015); Phys Rev. B 93, 140409 (2016); Nat. Nanotechnol. 9, 891 (2014)]. However, in our case, we have observed that the additional peak (S_Q/Se_Q) does not follow the quadratic behavior; instead, it exhibits a similar screening effect to the neutral exciton (Fig. 3). This deviation from the expected power law again supports that the assignment of the additional peak is not biexciton.
- v. **Absence in monolayer:** The absence of S_Q/Se_Q peaks in monolayer, even at higher laser power, and presence only on heterostructure provides additional evidence to rule out their association with biexciton.

In the previous response letter, we discussed how we have ruled out the possibilities for localized exciton, and therefore for the remainder, we repeat our explanation.

Localized exciton:

- i. **Flat clean surface:** We have observed a strong K-Q peak on a flat bubble-free clean surface (Supplementary Fig. 13), ensuring this peak is not related to any localized related defect-induced exciton.
- ii. **The indirect peak at intact bilayer:** Intact homogeneous bilayers, which exhibit no bubbles or residue at the interface, show similar peaks (Fig.1 g,h) to those observed in our heterostructure (Fig. 1a-f).
- iii. **Post-annealing effect:** Nanobubbles can be removed by post-annealing as previously reported by several groups including the authors of the reference paper group (2D mater, 2018, 5, 031001; 2D mater, 2017, 4, 021019). In our case, (i) we fabricated our heterostructure at a high temperature ($>110^\circ\text{C}$) and (ii) further post-annealing for 12 hours at 250°C . As- fabricated samples may contain nanobubbles and polymer contaminations, whereas polymer contaminations and nanobubbles are largely removed after post-annealing. The new peak S_Q/Se_Q remains intact before and after annealing, as shown in supplementary Fig. S17. This again confirms that this additional peak is not related to nanobubble. Furthermore, a strong S_Q peak was observed on a clean interface compared to the contaminated interface as discussed in Supplementary Fig.S4.

We appreciate the reviewer for raising these concerns, as they allowed us to further scrutinize and validate our findings.

Reviewer #3 (Remarks to the Author):

General comment:

The authors has provided a very thorough and extensive reply. They have provided additional data to clarify their interpretation that the observed signal in PL originates from an indirect exciton. They clearly put a lot of effort in their reply and revised manuscript which significantly improves its clarity with a better presentation of the novelty from their finding. Therefore, I think their revised manuscript is worth publishing in Nature Communications.

Response: We sincerely thank the reviewer for the positive feedback and for acknowledging the thoroughness and extensive effort we put into our response and revised the manuscript. We are delighted that our additional data and clarifications have contributed to a better interpretation of our findings. Thank you once again for your evaluation and for considering our revised manuscript for publication.

Reviewers' Comments:

Reviewer #2:

Remarks to the Author:

The authors add new interpretations on supporting their claims and I have no further comments.

REVIEWERS' COMMENTS

Reviewer #2 (Remarks to the Author):

General comment:

The authors add new interpretations on supporting their claims and I have no further comments.

Response: We sincerely thank the reviewer for carefully reviewing our manuscript. Reviewer's feedback and expertise have been incredibly valuable throughout this process in enhancing the quality of our work.